# Vernalization-triggered expression of the antisense transcript *COOLAIR* is mediated by *CBF* genes

Myeongjune Jeon[1,2†], Goowon Jeong[1,2†], Yupeng Yang[3], Xiao Luo[4], Daesong Jeong[1,2], Jinseul Kyung[1,2], Youbong Hyun[1,2], Yuehui He[3,4]*, Ilha Lee[1,2]*

[1]School of Biological Sciences, Seoul National University, Seoul, Republic of Korea; [2]Research Center for Plant Plasticity, Seoul National University, Seoul, Republic of Korea; [3]Shanghai Center for Plant Stress Biology & National Key Laboratory for Plant Molecular Genetics, CAS Center for Excellence in Molecular Plant Sciences, Chinese Academy of Sciences, Shanghai, China; [4]Peking University Institute of Advanced Agricultural Sciences, Weifang, China

*For correspondence:
yhhe@pku.edu.cn (YH);
ilhalee@snu.ac.kr (IL)

†These authors contributed equally to this work

**Abstract** To synchronize flowering time with spring, many plants undergo vernalization, a floral-promotion process triggered by exposure to long-term winter cold. In *Arabidopsis thaliana*, this is achieved through cold-mediated epigenetic silencing of the floral repressor, *FLOWERING LOCUS C* (*FLC*). *COOLAIR*, a cold-induced antisense RNA transcribed from the *FLC* locus, has been proposed to facilitate *FLC* silencing. Here, we show that C-repeat (CRT)/dehydration-responsive elements (DREs) at the 3′-end of *FLC* and CRT/DRE-binding factors (CBFs) are required for cold-mediated expression of *COOLAIR*. CBFs bind to CRT/DREs at the 3′-end of *FLC*, both in vitro and in vivo, and CBF levels increase gradually during vernalization. Cold-induced *COOLAIR* expression is severely impaired in *cbfs* mutants in which all *CBF* genes are knocked-out. Conversely, *CBF*-overexpressing plants show increased *COOLAIR* levels even at warm temperatures. We show that *COOLAIR* is induced by CBFs during early stages of vernalization but *COOLAIR* levels decrease in later phases as *FLC* chromatin transitions to an inactive state to which CBFs can no longer bind. We also demonstrate that *cbfs* and *FLCΔCOOLAIR* mutants exhibit a normal vernalization response despite their inability to activate *COOLAIR* expression during cold, revealing that *COOLAIR* is not required for the vernalization process.

## Editor's evaluation

This important work advances our understanding of the systems that plants evolved to coordinate developmental processes such as the timing of flowering with seasonal change, particularly with respect to the regulation and role of long non-coding RNAs (lncRNAs) complementary to genes encoding proteins that regulate developmental switches. The evidence supporting the conclusions is solid. The work will be of interest to those interested plant development as well those interested in the role and regulation of lncRNAs.

## Introduction

Appropriate timing of flowering provides evolutionary advantage of plant reproductive success. As sessile organisms, plants have evolved mechanisms through which seasonal cues coordinate the transition to flowering. One of the significant environmental factors affecting flowering time of plants adapted to temperate climates is the temperature changes during the seasons, and plants have

**eLife digest** Long spells of cold winter weather may feel miserable, but they are often necessary for spring to blossom. Indeed, many plants need to face a prolonged period of low temperatures to be able to flower; this process is known as vernalization.

While the molecular mechanisms which underpin vernalization are well-known, it is still unclear exactly how plants can 'sense' the difference between short and long periods of cold. Jeon, Jeong et al. set out to explore this question by focusing on *COOLAIR*, one of the rare genetic sequences identified as potentially being able to trigger vernalization. *COOLAIR* is a long noncoding RNA, a partial transcript of a gene that will not be 'read' by the cell to produce a protein but which instead regulates how and when certain genes are being switched on. *COOLAIR* emerges from the locus of the *FLC* gene, which is one of the main repressors of flowering, and it gradually accumulates in the plant when temperatures remain low for a long period. While some evidence suggests that *COOLAIR* may help to switch off *FLC*, other studies have raised some doubts about its involvement in vernalization.

In response, Jeon, Jeong et al. examined the *FLC* gene in a range of plants closely related to *A. thaliana*, and in which *COOLAIR* also accumulates upon cold exposure. This helped them identify a class of proteins, known as CBFs, which could bind to sequences near the *FLC* gene to activate the production of *COOLAIR* when the plants were kept in cold conditions for a while. CBFs were already known to help plants adapt to short cold snaps, but these experiments confirmed that they could act as both short- and long-term cold sensors.

This work allowed Jeon, Jeong et al. to propose a model in which CBF and therefore *COOLAIR* levels increase as the cold persists, until changes in the structure of the *FLC* gene prevent CBF from binding to it and *COOLAIR* production drops. Unexpectedly, examining the fate of mutants which could not produce *COOLAIR* revealed that these plants could still undergo vernalization, suggesting that the long noncoding RNA is in fact not necessary for this process.

These results should prompt other scientists to further investigate the role of *COOLAIR* in vernalization; they also give insight into how coding and noncoding sequences may have evolved together in various members of the *A. thaliana* family to adapt to the environment.

evolved complex sensory mechanisms to monitor the surrounding temperature to properly control the timing of flowering and tolerate thermal stress (*Went, 1953*; *Penfield, 2008*; *Ding and Yang, 2022*).

Cold acclimation and vernalization are two responses of plants to low temperatures. Cold acclimation is generally initiated by a short period of non-freezing cold exposure and increases the frost tolerance of plants (*Weiser, 1970*; *Gilmour et al., 1988*; *Guy, 1990*; *Thomashow, 1999*; *Chinnusamy et al., 2007*). The three C-REPEAT (CRT)/DEHYDRATION-RESPONSIVE ELEMENT (DRE) BINDING FACTORs (CBFs) and their encoding genes serve as signaling hubs for cold acclimation in *Arabidopsis thaliana* (*Stockinger et al., 1997*; *Medina et al., 1999*; *Thomashow, 2010*; *Ding et al., 2019*). When exposed to low temperatures, the transcription of *CBFs* is rapidly promoted by a group of cold-signal transducers, including the $Ca^{2+}$/calmodulin-binding proteins, CALMODULIN-BINDING TRANSCRIPTION ACTIVATORs (CAMTAs) (*Doherty et al., 2009*; *Kim et al., 2013*; *Kidokoro et al., 2017*); the clock proteins, CIRCADIAN CLOCK-ASSOCIATED 1 (CCA1) and LATE-ELONGATED HYPOCOTYL (LHY) (*Dong et al., 2011*); and the brassinosteroid-responsive proteins, BRASSINAZOLE-RESISTANT 1 (BZR1) and CESTA (CES) (*Eremina et al., 2016*; *Li et al., 2017*). In addition, cold also enhances the stability or activity of CBF proteins. For example, cold facilitates the interaction between CBFs and BASIC TRANSCRIPTION FACTOR 3s (BTF3s), which promotes CBF stability (*Ding et al., 2018*), and cold triggers degradation of co-repressor, HISTONE DEACETYLASE 2C (HD2C), thereby allowing CBFs to activate their targets (*Park et al., 2018b*). Furthermore, cold reduces oxidized CBFs, which increases active CBF monomers (*Lee et al., 2021*). Low-temperature-induced CBFs, in turn, activate the expression of *cold-regulated* (*COR*) genes by binding to CRT/DREs in their promoters (*Stockinger et al., 1997*; *Medina et al., 1999*). Diverse arrays of cryoprotective proteins encoded by *COR* genes allow plants to overcome freezing stress (*Gilmour et al., 1988*; *Hajela et al., 1990*; *Thomashow et al., 1997*; *Shinozaki et al., 2003*; *Shi et al., 2018*; *Ding et al., 2019*).

In contrast to cold acclimation, vernalization, a floral-promotion process that occurs during winter, requires an extended cold period (*Napp-Zinn, 1955*; *Chouard, 1960*; *Michaels and Amasino, 2000*).

This allows plants to synchronize the timing of flowering with favorable spring conditions. Vernalization in *A. thaliana* is mainly achieved by silencing the floral repressor gene, *FLOWERING LOCUS C* (*FLC*) (*Michaels and Amasino, 1999*; *Sheldon et al., 1999*; *Sheldon et al., 2000*; *Michaels and Amasino, 2001*). *FLC* encodes a MADS-box protein that represses the expression of floral activator genes, *SUPPRESSOR OF OVEREXPRESSION OF CO 1* (*SOC1*) and *FLOWERING LOCUS T* (*FT*, encoding florigen), by directly binding to their promoter regions (*Lee et al., 2000*; *Michaels et al., 2005*; *Helliwell et al., 2006*). In *Arabidopsis* winter annuals, such as San Feliu-2, Löv-1, and Sweden (SW) ecotypes, flowering is prevented by the high expression of *FLC* before exposure to winter cold (*Lee et al., 1993*; *Shindo et al., 2005*; *Park et al., 2018a*). This is caused by the strong transcriptional activation of *FLC* by the FRIGIDA (FRI) supercomplex, which recruits general transcription factors and several chromatin modifiers (*Michaels and Amasino, 1999*; *Sheldon et al., 1999*; *Johanson et al., 2000*; *Choi et al., 2011*; *Li et al., 2018*). Prior to vernalization, *FLC* chromatin is highly enriched with active histone marks such as histone H3 acetylation and trimethylation of Lys4 or Lys36 at H3 (H3K4me3/H3K36me3) (*Bastow et al., 2004*; *Yang et al., 2014*). In contrast, prolonged cold exposure results in gradual deacetylation of *FLC* chromatin and concomitant removal of H3K4me3 and H3K36me3 from the *FLC* (*Bastow et al., 2004*; *Yang et al., 2014*; *Nishio et al., 2016*). Additionally, VP1/ABI3-LIKE 1 (VAL1) and VAL2 recruit Polycomb Repressive Complex 2 (PRC2) onto *FLC* chromatin, thereby accumulating the repressive histone mark, H3 Lys27 trimethylation (H3K27me3), in the nucleation region around the first exon and intron of *FLC* (*Sung and Amasino, 2004b*; *Wood et al., 2006*; *De Lucia et al., 2008*; *Angel et al., 2011*; *Nishio et al., 2016*; *Qüesta et al., 2016*; *Yuan et al., 2016*). Subsequently, upon returning to warm temperatures, H3K27me3 marks are spread over the entire *FLC* chromatin region by LIKE HETEROCHROMATIN PROTEIN 1 (LHP1), which ensures stable *FLC* suppression and renders plants competent to flower (*Mylne et al., 2006*; *Sung et al., 2006*; *Yang et al., 2017*).

Several long-term cold-induced factors have been shown to play crucial roles in the epigenetic silencing of *FLC. VERNALIZATION INSENSITIVE 3* (*VIN3*) family genes, which are upregulated by prolonged cold, encode plant homeodomain (PHD) proteins that recognize H3K9me2 enriched in *FLC* chromatin during vernalization (*Sung and Amasino, 2004b*; *Kim and Sung, 2013*; *Kim and Sung, 2017a*). These proteins mediate the recruitment of PRC2 and the subsequent deposition of H3K27me3 at the *FLC* nucleation region (*Sung and Amasino, 2004b*; *De Lucia et al., 2008*; *Kim and Sung, 2013*). In addition, vernalization-induced long noncoding RNAs (lncRNAs) are involved in such histone modifications. *COLDAIR* and *COLDWRAP*, the two lncRNAs transcribed from the first intron and promoter region of *FLC*, respectively, are required for H3K27me3 deposition in response to long-term cold (*Heo and Sung, 2011*; *Kim and Sung, 2017b*). *COLDAIR* and *COLDWRAP* are also thought to affect the formation of the intragenic chromatin loop at the *FLC*, which may be a part of the *FLC* silencing mechanism (*Kim and Sung, 2017b*). Unlike *COLDAIR* and *COLDWRAP*, which are transcribed in the sense direction of *FLC*, another lncRNA, *COOLAIR,* is an antisense transcript expressed from the 3′-end of *FLC* (*Swiezewski et al., 2009*). The gradual accumulation of *COOLAIR* reaches a maximum within a few weeks of cold exposure, whereas *COLDAIR* and *COLDWRAP* show peaks at a later phase of vernalization (*Csorba et al., 2014*; *Kim and Sung, 2017b*). *COOLAIR* was reported to remove active histone marks from *FLC* chromatin (*Liu et al., 2007*; *Csorba et al., 2014*; *Fang et al., 2020*; *Xu et al., 2021a*; *Zhu et al., 2021*). Particularly, in summer annuals, phase-separated RNA-processing complexes favor co-transcriptional proximal polyadenylation of *COOLAIR* (*Marquardt et al., 2014*; *Wang et al., 2014*; *Fang et al., 2019*; *Wu et al., 2020*; *Xu et al., 2021b*). The *COOLAIR*-processing machinery exhibits transient and dynamic interactions with an H3K4 demethylation complex, leading to *FLC* suppression at warm temperatures (*Liu et al., 2007*; *Fang et al., 2020*; *Xu et al., 2021a*). *COOLAIR* is also likely to be involved in reducing H3K36me3 at the *FLC* during vernalization process (*Csorba et al., 2014*). *COOLAIR* was reported to promote the sequestration of the FRI complex from the *FLC* promoter by condensing it into phase-separated nuclear bodies (*Zhu et al., 2021*). This has been suggested to cause the inactivation of *FLC,* which is probably accompanied by the silencing of *FLC* chromatin through the removal of H3K36me3. However, a previous study has raised the issue that *COOLAIR* appears not to be necessary for vernalization (*Helliwell et al., 2011*; *Luo et al., 2019*).

Compared with the signaling pathway of cold acclimation, how a long-term cold signal is transduced to trigger the induction of *VIN3* and lncRNAs is less well understood. It is marginally known that some chromatin modifiers, NAC WITH TRANSMEMBRANE MOTIF 1-LIKE 8 (NTL8), and CCA1/LHY

act as positive regulators of *VIN3* during the vernalization process (*Kim et al., 2010*; *Jean Finnegan et al., 2011*; *Zhao et al., 2020*; *Kyung et al., 2022*). However, little is known about the upstream regulators of *COOLAIR* required for cold-induction. Recent reports have shown that an NAC domain-containing protein, NTL8, and the WRKY transcription factor, WRKY63, can bind to the promoter of *COOLAIR* and activate its expression (*Zhao et al., 2021*; *Hung et al., 2022*). However, whether NTL8 and WRKY63 are necessary for the full extent of *COOLAIR* induction during vernalization has not been thoroughly addressed. In this study, we identified that CRT/DREs at the 3′-end of the *FLC* are required for the long-term cold response of *COOLAIR*. Additionally, we show that CBFs, which accumulate during the long-term winter cold, act as upstream regulators of *COOLAIR* during vernalization.

## Results

### A CRT/DRE-binding factor, CBF3, directly binds to CRT/DREs at the 3′-end of the *FLC*

The proximal promoter region of *COOLAIR* is highly conserved among *FLC* orthologs from *A. thaliana* relatives (*Castaings et al., 2014*). Thus, we assumed that the cis-element conferring a long-term cold response would exist within that block. A comparison of the region near the transcriptional start site (TSS) of *COOLAIR* revealed the conservation of two CRT/DREs among six *FLC* orthologs from five related species of the Brassicaceae family (*Figure 1A*). CRT/DRE, the core sequence of which is CCGAC, is a regulatory element that imparts cold- or drought-responsive gene expression (*Baker et al., 1994*). *CBF1*, *2*, and *3* encode APETALA 2 (AP2) domain-containing proteins that can bind to CRT/DRE and are usually present in the promoters of cold- and drought-responsive genes (*Stockinger et al., 1997*; *Medina et al., 1999*). Consistent with this, the *Arabidopsis* cistrome database and the genome-wide chromatin immunoprecipitation sequencing (ChIP-seq) results from a previous study suggested that CBFs bind to the 3′-end sequence of *FLC* containing CRT/DREs (*Figure 1—figure supplement 1*; *O'Malley et al., 2016*; *Song et al., 2021*).

We performed an electrophoretic mobility shift assay (EMSA) to confirm this binding using probes harboring CRT/DREs (named DRE1 or 2) from the *COOLAIR* promoter. The mobility of these two Cy5-labeled probes was retarded by maltose-binding protein (MBP)-fused CBF3 (*Figure 1B*, *Figure 1—figure supplement 2*). The band shift was competed out by adding an excess amount of unlabeled DRE1 or 2 oligonucleotides. In contrast, competitors containing mutant forms of DRE1 or 2 (DRE1$^m$ or 2$^m$, respectively) failed to compete (*Figure 1B*). We also tested the binding of other CRT/DRE-like sequences in the *FLC* locus. Two (DREa and b) were present in the *FLC* promoter, while the other two (DREc and d), were present in the first and last exons, respectively (*Figure 1—figure supplement 2*). Only DREc competed with the band shift caused by the CBF3-DRE1 interaction (*Figure 1B*, *Figure 1—figure supplement 2*). The presence or absence of bases that determine the binding affinity between CBFs and CRT/DRE could explain the differences in CBF3-binding patterns among CRT/DRE-like sequences at the *FLC* locus (*Maruyama et al., 2004*).

Subsequently, a chromatin immunoprecipitation (ChIP) assay using a *CBF3*-overexpressing transgenic plant, *pSuper:CBF3-myc* (*Liu et al., 2017*), was conducted to determine whether CBF3 is associated with the *FLC* region containing CRT/DREs in vivo. For ChIP assay to show binding affinity of CBF3 protein to the *COOLAIR* promoter during vernaliation, we used *pSuper:CBF3-myc* instead of *pCBF3:CBF3-myc* because endogenous *CBF3* transcript level is changed during vernalization as shown afterward. Consistent with the in vitro results, CBF3-myc protein was enriched at the 3′-end of *FLC* and the first exon where DREc was located (*Figure 1C*). Without vernalization (NV), CBF3-myc was highly enriched at both the 5′- and 3′-ends of *FLC* (P1 and P3), indicating that CBF3 protein can activate *FLC* under warm conditions, as previously reported (*Seo et al., 2009*). However, enrichment in the P1 region rapidly disappeared during the vernalization period. In contrast, enrichment on P3 was maintained until 10 days of vernalization (10V) and was subsequently reduced. This is coincident with the expression pattern of *COOLAIR*, which declines during the late phase of vernalization (*Csorba et al., 2014*).

### *COOLAIR* is one of the CBF regulons in *Arabidopsis*

*CBF* genes are rapidly and transiently induced upon exposure to cold (*Medina et al., 1999*). CBF regulons, which are CBF-targeted genes, are correspondingly up- or down-regulated after cold intervals

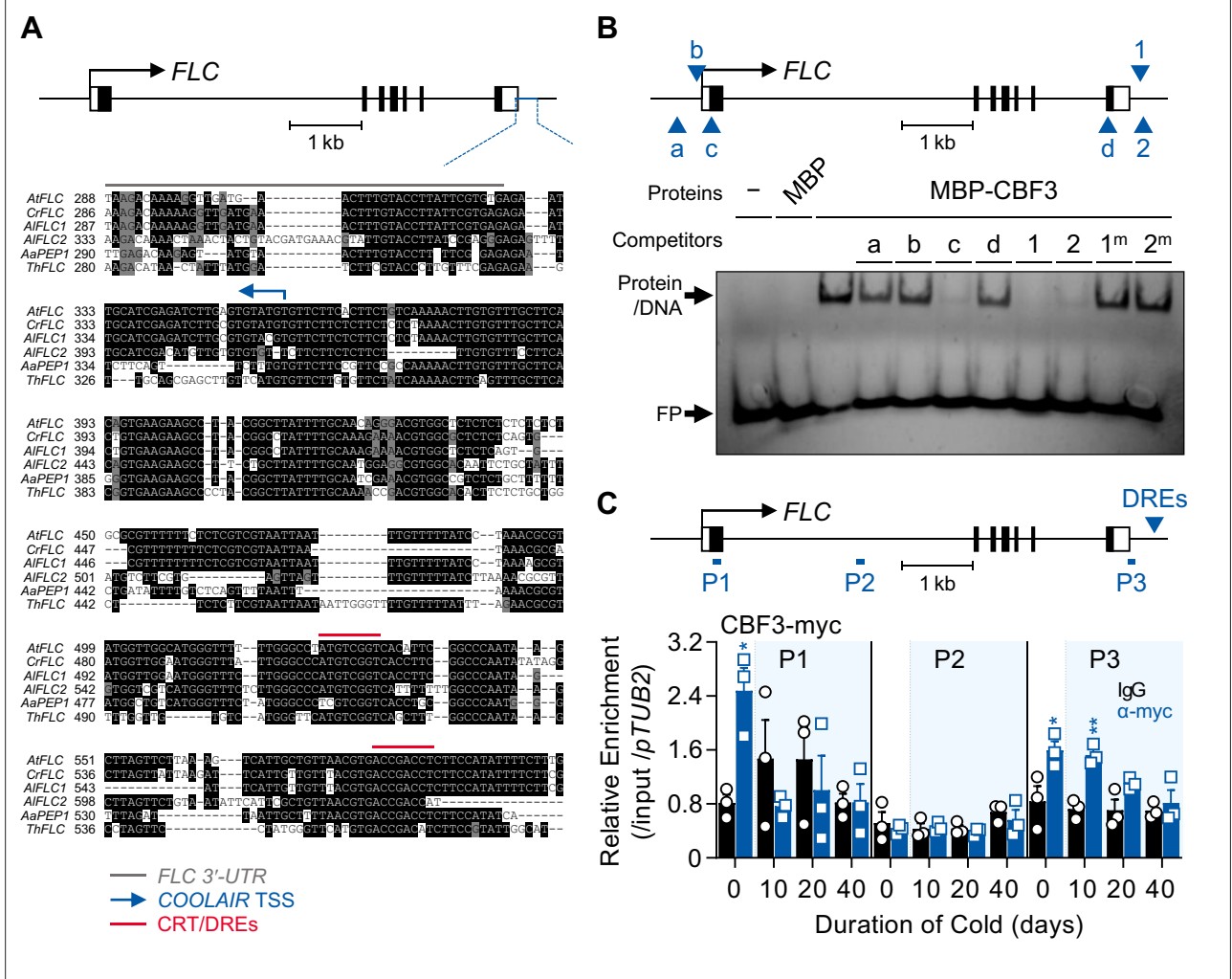

**Figure 1.** CBF3 directly binds to the CRT/DREs at the 3'-end of *FLC*. (**A**) Comparison of sequences around the 3'-end of the *FLC* orthologs from *Arabidopsis* relatives. The upper graphic presents the gene structure of *AtFLC*. The black bars, black lines, and white bars indicate exons, introns, and untranslated regions (UTRs), respectively. The blue line presents the region used for sequence comparison among six orthologous genes from five plant species. In the sequence alignment below, the gray line indicates the 3'-UTR of *FLC* orthologs, the blue arrow indicates the transcriptional start site (TSS) of *AtCOOLAIR*, and the red lines indicate CRT/DREs. At, *Arabidopsis thaliana*; Cr, *Capsella rubella*; Al, *Arabidopsis lyrata*; Aa, *Arabis alpina*; Th, *Thellungiella halophila*. (**B**) EMSA using one of the CRT/DREs located at the *AtCOOLAIR* promoter, DRE1, as a probe. In the upper graphic showing *AtFLC* gene structure, CRT/DRE-like sequences are marked as blue arrows and labeled as a, b, c, and d for CRT/DRE-like sequences and as 1 and 2 for CRT/DRE sequences. For the competition assay, these CRT/DRE-like sequences and mutant forms of DRE1 and DRE2 were used as competitors. A 100-fold molar excess of unlabeled competitors was added. No protein (−) or maltose-binding protein (MBP) were used as controls. FP, free probe. (**C**) ChIP assay result showing the enrichment of CBF3-myc protein on the *AtFLC* locus. Samples of NV, 10V, 20V, and 40V plants of *pSuper:CBF3-myc* were collected at zeitgeber time (ZT) 4 in an SD cycle. The CBF3-chromatin complex was immunoprecipitated (IP) with anti-myc antibodies (blue bars), and mouse IgG (black bars) was used as a control. Positions of qPCR amplicons used for ChIP-qPCR analysis are illustrated as P1, P2, and P3 in the upper graphic. The blue arrow in the graphic denotes the position of CRT/DREs on the *AtCOOLAIR* gene. ChIP-qPCR results have been represented as mean ± SEM of the three biological replicates in the lower panel. Open circles and squares represent each data point. Relative enrichments of the IP/5% input were normalized to that of *pTUB2*. The blue shadings indicate cold periods. Asterisks indicate a significant difference between IgG and anti-myc ChIP-qPCR results at each vernalization time point (*, $p < 0.05$; **, $p < 0.01$; unpaired Student's *t*-test).

The online version of this article includes the following source data and figure supplement(s) for figure 1:

**Source data 1.** Uncropped labeled gel image and the original image file for the EMSA result.

**Figure supplement 1.** CBF proteins directly bind to the sequence at the 3'-end of *FLC* in DAP-seq.

**Figure supplement 2.** CBF3 binds to DRE1 and DRE2 on the *COOLAIR* promoter and DREc on the first exon of *FLC*.

**Figure supplement 2—source data 1.** Uncropped labeled gel images and the original image files for the EMSA results.

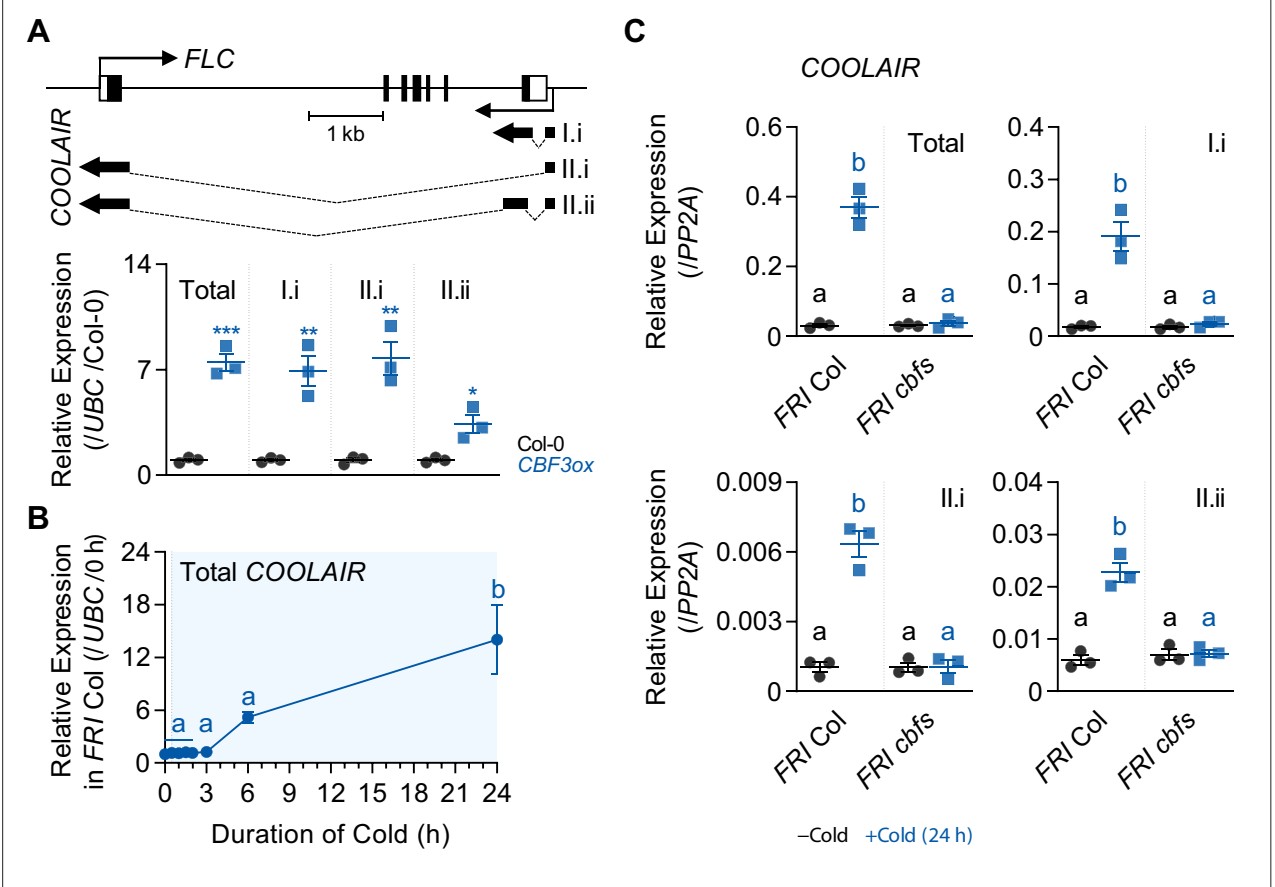

**Figure 2.** *COOLAIR* shows a similar expression pattern with CBF regulons. (**A**) Expression levels of total *COOLAIR* and *COOLAIR* isoforms in the wild-type (Col-0) and *CBF3*-overexpressing plants (*pSuper:CBF3-myc* [*CBF3ox*]). The gene structures of *FLC* and *COOLAIR* variants are illustrated in the upper panel. The thin black arrows indicate the transcription start sites of *FLC* and *COOLAIR*. The thick black arrows indicate the exons of each *COOLAIR* isoform. The structures of proximal (I.i) and distal (II.i and II.ii) *COOLAIR* isoforms are shown. The primer sets used for total, proximal (I.i), and distal (II.i, II.ii) *COOLAIR*s are marked in *Figure 2—figure supplement 1*. Relative transcript levels of total *COOLAIR* and *COOLAIR* variants were normalized to that of *UBC* and have been represented as mean ± SEM of three biological replicates. Dots and squares represent each data point. Asterisks indicate a significant difference as compared to the wild type (*, $p < 0.05$; **, $p < 0.01$; ***, $p < 0.001$; unpaired Student's *t*-test). (**B**) Expression dynamics of total *COOLAIR* after short-term (0, 0.5, 1, 1.5, 2, 3, 6, and 24 hr) cold treatment. Wild types (*FRI* Col) were subjected to 4 °C cold and harvested at each time point. Relative transcript levels of total *COOLAIR* to *UBC* were normalized to that of non-cold treated wild type. The values have been represented as mean ± SEM of three biological replicates. The blue shading indicates periods under cold treatment. Significant differences have been marked using different letters (a, b; $p < 0.05$; one-way ANOVA followed by Tukey's post-hoc test). (**C**) Transcript levels of total *COOLAIR* and *COOLAIR* isoforms in wild type and *cbfs-1* mutant before (−Cold) and after (+Cold) a day of 4 °C cold treatment. Relative levels of total *COOLAIR* and *COOLAIR* variants were normalized to that of *PP2A*. Values have been represented as mean ± SEM of three biological replicates. Dots and squares indicate each data point. Significant differences have been marked using different letters (a, b; $p < 0.05$; two-way ANOVA followed by Tukey's post-hoc test).

The online version of this article includes the following figure supplement(s) for figure 2:

**Figure supplement 1.** Overexpression of *CBF1*, *2*, or *3* causes an increase in *COOLAIR* level, but the effects on *COOLAIR* variants are subtly different.

of a few hours (*Gilmour et al., 1998*; *Liu et al., 1998*; *Fowler and Thomashow, 2002*). As *COOLAIR* contains CBF3-binding sites in its promoter (*Figure 1*), we analyzed whether *COOLAIR* shows CBF regulon-like expression. The expression of genes assigned to the CBF regulon is up- or down-regulated by *CBF* overexpression under warm conditions (*Park et al., 2015*). Similarly, we found that the total transcript level of *COOLAIR* increased in *CBF3*-overexpressing transgenic plants (*pSuper:CBF3-myc*) grown at room temperature (22 °C) compared to that in the wild type, Columbia-0 (Col-0) (*Figure 2A*). Such upregulation was mainly due to the type I.i or type II.i *COOLAIR* isoforms. As the levels of both proximal (type I) and distal (type II) variants of *COOLAIR* were higher in the *CBF3* overexpressor than in the wild type, it is likely that *COOLAIR* transcription, instead of the 3′-processing events, is affected

by CBF3 (*Figure 2A*; *Liu et al., 2010*; *Marquardt et al., 2014*). It has been reported that the targets of CBF1 or 2 are not entirely identical to those of CBF3, although *CBF* genes show high sequence similarity (*Novillo et al., 2007*). Thus, we compared the expression levels of all *COOLAIR* isoforms in transgenic plants overexpressing *CBF1*, *2*, or *3* (*Seo et al., 2009*). Overall, all CBF-overexpressing plants showed increased levels of *COOLAIR* (*Figure 2—figure supplement 1*). However, each *CBF*-overexpressing plant showed a subtly different effect on *COOLAIR* expression. The *CBF2* overexpressor showed the highest level of the proximal *COOLAIR* isoform, whereas the *CBF3* overexpressor showed the highest level of distal *COOLAIR*. In contrast, the *CBF1*-overexpressing plants showed a slightly smaller increase in proximal *COOLAIR* levels and a negligible increase in distal *COOLAIR*, compared to those in the wild type, Wassilewskija-2 (Ws-2). Such results imply that the three CBFs regulate *COOLAIR* transcription subtly differently.

Another characteristic of CBF regulons is that their expression is activated by a day of cold exposure (*Park et al., 2015*). Thus, we analyzed if *COOLAIR* is also induced by short-term cold exposure. We treated wild-type (*FRI* Col) plants with 0, 0.5, 1, 1.5, 2, 3, 6, and 24 hr of cold (4 °C), then measured the levels of total *COOLAIR*. The results showed that cold treatment for less than 3 hr failed to induce *COOLAIR* expression, although *CBF* transcripts reached a peak at 3 hr of cold exposure (*Figures 2B and 3A*; *Gilmour et al., 1998*). However, after 6 hr of cold treatment, when CBF3 protein levels peaked, *COOLAIR* was strongly induced (*Figures 2B and 3D*). Thus, short-term cold-triggered *COOLAIR* expression dynamics were similar to those of other CBF-targeted genes; that is, the expression was highly increased after *CBF* transcript levels reached a peak upon cold exposure (*Gilmour et al., 1998*; *Fowler and Thomashow, 2002*). To confirm whether *CBFs* are responsible for the rapid cold response of *COOLAIR*, we examined *COOLAIR* induction after a day of cold treatment in the wild type and *cbfs-1* mutant, in which all three *CBFs* were knocked out (*Figure 2C*; *Jia et al., 2016*). All isoforms, as well as total *COOLAIR*, failed to be induced by short-term cold exposure in the *cbfs* mutant, although a drastic increase was observed in the wild type. Thus, these results strongly suggest that *CBFs* are required for *COOLAIR* induction, even under a short period of cold exposure.

## CBFs accumulate during vernalization

We subsequently investigated whether *CBFs* are also responsible for vernalization-induced *COOLAIR* activation. It has been reported that *COOLAIR* is gradually upregulated as the cold period persists and peaks after 2–3 weeks of vernalization (*Csorba et al., 2014*). However, most studies on *CBF* expression have been performed within a few days since the function of *CBFs* has been analyzed only in the context of short-term cold (*Gilmour et al., 1998*; *Liu et al., 1998*; *Medina et al., 1999*; *McKhann et al., 2008*). Therefore, we investigated the expression patterns of *CBFs* before and after long-term cold exposure to determine the correlation between the expression of *CBFs* and *COOLAIR* during vernalization. As previously shown, the levels of all three *CBFs* peaked within 3 hr of cold exposure and then decreased rapidly (*Figure 3A*). However, *CBF* levels increased again as the cold period was prolonged (> 20V), suggesting that both short-term and vernalizing cold treatments upregulated *CBF* expression (*Figure 3B*).

Because *CBFs* exhibit rhythmic oscillations (*Dong et al., 2011*), we also investigated whether the rhythmic expression of *CBFs* is affected by vernalization. Wild-type (Col-0) plants were collected every 4 hr under a short-day photoperiod (SD, 8 hr light/16 hr dark) under both NV and 40V conditions. As shown in *Figure 3C*, although the rhythmic pattern of each *CBF* was variable, the overall transcript levels of all three *CBFs* were much higher at 40V than under NV. Consistent with the transcript level, the level of CBF3 protein in vernalized plants was higher than that in NV plants during all circadian cycles (*Figure 3—figure supplement 1*).

To verify whether the increased transcription of *CBF3* led to protein accumulation, we measured the level of CBF3 at each time point of cold treatment using the *pCBF3:CBF3-myc* plant (*Jia et al., 2016*). Following the rapid and transient induction of *CBF3* under short-term cold conditions (*Figure 3A*; *Medina et al., 1999*), CBF3-myc protein levels peaked within 6 hr of cold exposure and then decreased (*Figure 3D*). Thus, the peak of the *CBF3* transcript and that of the CBF3 protein showed a gap of few hours. Notably, the peak of the CBF3 protein correlated with the time when *COOLAIR* was rapidly induced (*Figures 2B and 3D*). During vernalization, the level of CBF3-myc protein gradually declined until 10V, but then increased again (*Figure 3E*). Thus, the level at 40V is similar to that at 1V. When the plants were transferred to room temperature for 4 d after 40V (40VT4),

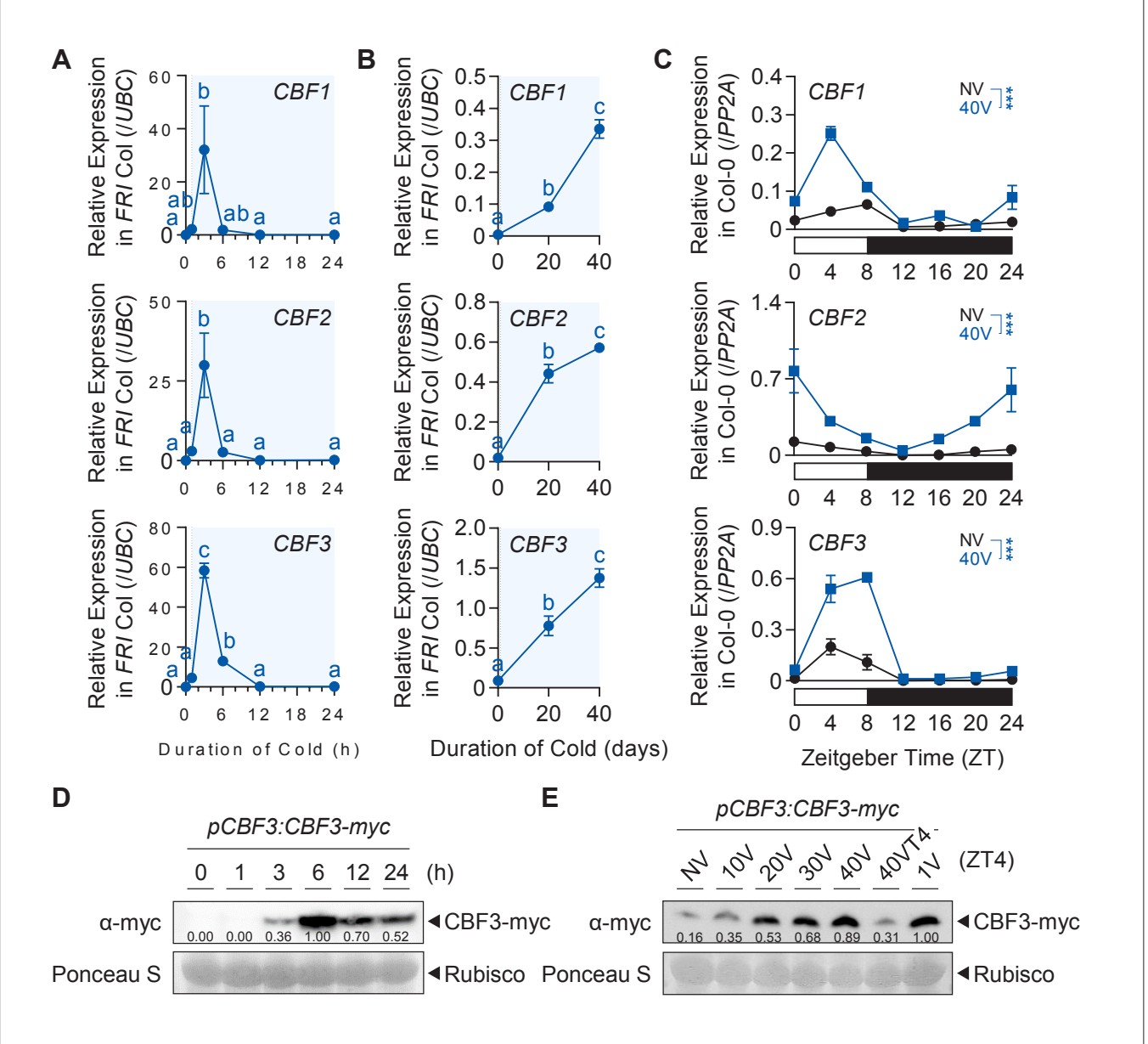

**Figure 3.** Protein levels of CBFs increase during the vernalization process. (**A**) Transcript levels of *CBF1*, *2*, and *3* under short-term cold exposure. Wild-type plants were subjected to 0, 1, 3, 6, 12, and 24 hr of 4 °C cold treatment. Relative levels of *CBF1*, *2*, and *3* were normalized to that of *UBC*. Values have been represented as mean ± SEM of three biological replicates. The blue shadings indicate cold periods. Significant differences have been marked using different letters (a–c; $p < 0.05$; one-way ANOVA followed by Tukey's post-hoc test). (**B**) Transcript levels of *CBF1*, *2*, and *3* after 20V and 40V. Wild-type plants were treated with 4 °C vernalization under an SD cycle and collected at ZT4. Relative levels of *CBF1*, *2*, and *3* were normalized to that of *UBC*. Values have been represented as mean ± SEM of three biological replicates. The blue shadings denote periods under cold. Significant differences have been marked using different letters (a–c; $p < 0.05$; one-way ANOVA followed by Tukey's post-hoc test). (**C**) Daily rhythms of *CBF1*, *2*, and *3* transcript levels in NV or 40V plants. Col-0 plants grown under an SD cycle were collected every 4 hr between ZT0 and ZT24. Relative transcript levels of *CBF1*, *2*, and *3* were normalized to that of *PP2A*. Values have been represented as mean ± SEM of three biological replicates. The white and black bars represent light and dark periods, respectively. Asterisks indicate a significant difference between NV and 40V (***, $p < 0.001$; two-way ANOVA). (**D**) Dynamics of CBF3 protein level under short-term cold exposure. The *pCBF3:CBF3-myc* transgenic plants were subjected to 0, 1, 3, 6, 12, and 24 hr of 4 °C cold treatment. CBF3 proteins were detected using anti-myc antibodies. Rubisco was considered the loading control. Numbers below each band indicate relative signal intensity compared to 6 hr. The mean values of two biological replicates are presented. (**E**) Increase of CBF3 protein level during the vernalization process. The *pCBF3:CBF3-myc* transgenic plants, subjected to 4 °C vernalization, were collected at ZT4 of the indicated time point. CBF3 proteins were detected using anti-myc antibodies. Rubisco was considered the loading control. Numbers below each band indicate relative signal intensity compared to 1V. The mean values of three biological replicates are presented.

*Figure 3 continued on next page*

Figure 3 continued

The online version of this article includes the following source data and figure supplement(s) for figure 3:

**Source data 1.** Uncropped labeled blot images and the original image files for the immunoblots.

**Figure supplement 1.** Vernalization increases CBF3 protein levels with a minor effect on the daily rhythm.

**Figure supplement 1—source data 1.** Uncropped labeled blot images and the original image files for the immunoblots.

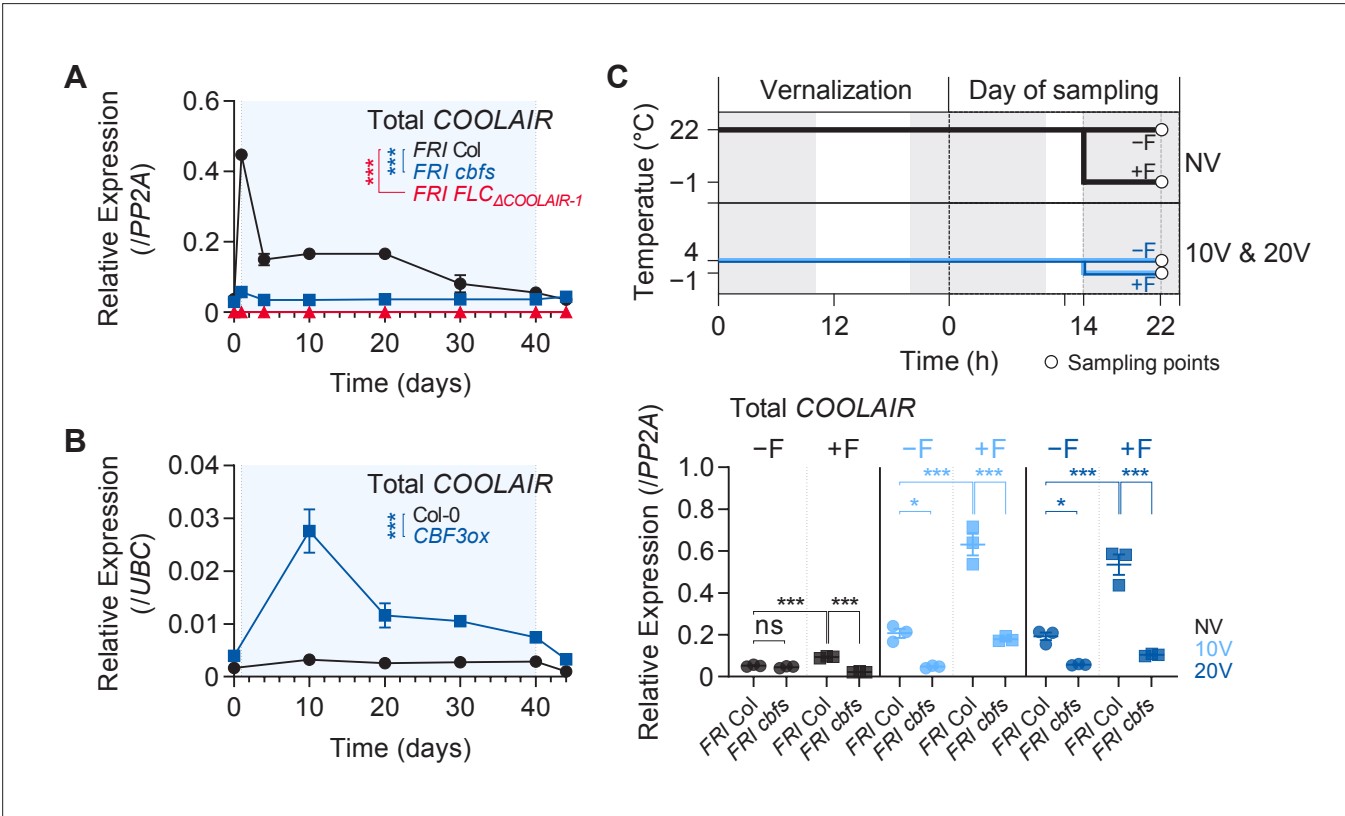

**Figure 4.** *CBF*s are involved in vernalization-induced *COOLAIR* expression. (**A**) Expression dynamics of total *COOLAIR* in the wild-type, *cbfs-1*, and *FLC_ΔCOOLAIR-1* plants during vernalization. Relative transcript levels of total *COOLAIR* were normalized to that of *PP2A*. Values have been represented as mean ± SEM of three biological replicates. Asterisks indicate a significant difference compared to the wild-type (***, p<0.001; two-way ANOVA followed by Tukey's post-hoc test). (**B**) Transcript level of total *COOLAIR* in wild type (Col-0) and *CBF3*-overexpressing transgenic plant (*pSuper:CBF3-myc* [*CBF3ox*]) during vernalization. Relative levels of total *COOLAIR* were normalized to that of *PP2A*. Values have been represented as mean ± SEM of three biological replicates. Asterisks indicate a significant difference between the wild-type and *CBF3* overexpressor (***, p<0.001; two-way ANOVA). Blue shading denotes cold periods in (**A**) and (**B**). (**C**) Effect of first frost-mimicking treatment (8 hr of freezing [<0 °C]) on the level of total *COOLAIR* in NV, 10V, and 20V wild type and *cbfs-1*. The upper panel shows a schematic of the experimental procedure. The non-frost treated (−F) wild type and *cbfs-1* were collected at ZT22 after an 8 hr of dark treatment at 22 °C (NV) or 4 °C (10V and 20V). For the first frost treatment, wild type and *cbfs-1* mutant were treated with an additional 8 hr of −1 °C (+F) under dark, and then the whole seedlings were collected at ZT22 for analysis. All the plants were grown under an SD cycle. The gray shadings denote dark periods. Total *COOLAIR* levels have been represented as mean ± SEM of three biological replicates in the lower panel. Dots and squares indicate each data point. Relative levels of total *COOLAIR* were normalized to that of *PP2A*. Asterisks indicate a significant difference (*, p<0.05; ***, p<0.001; two-way ANOVA followed by Tukey's post-hoc test). ns, not significant (p≥0.05).

The online version of this article includes the following figure supplement(s) for figure 4:

**Figure supplement 1.** CBF regulons show strongly reduced expression in *cbfs* mutants similar to *COOLAIR*.

**Figure supplement 2.** Cold-triggered *COOLAIR* expression is impaired in *cbfs* mutants in Col-0 and Sweden-ecotype background.

**Figure supplement 3.** Freezing temperature increases the levels of *CBFs*.

the protein level rapidly decreased. These results suggest that CBFs are responsible for the progressive upregulation of *COOLAIR*, at least during the early phase of vernalization (*Csorba et al., 2014*).

## *CBFs* are involved in vernalization-induced *COOLAIR* expression

To clarify whether *CBFs* are responsible for long-term cold-mediated *COOLAIR* induction, we quantified *COOLAIR* levels in wild type and *cbfs* mutant during vernalization. As shown in *Figure 2B and C*, the *COOLAIR* level was considerably increased by a day of low-temperature treatment in the wild type but quickly declined afterward (*Figure 4A*). The *COOLAIR* level increased again after 4V and reached a secondary peak at approximately 10V to 20V. It then decreased during the remaining vernalization period, as similar to previous results (*Csorba et al., 2014*). The expression dynamics of *COOLAIR* were similar to those of two well-known CBF targets, *COR15A* and *RESPONSIVE TO DESICCATION 29*A (*RD29A*), although their levels remained high until the end of the vernalization period (*Figure 4—figure supplement 1*). In contrast to the wild-type plants, the *cbfs* mutant showed severely reduced *COOLAIR* expression, as well as *COR15A* and *RD29A* levels, during all the period of vernalization (*Figure 4A*, *Figure 4—figure supplement 1*). Analysis of RNA sequencing (RNA-seq) data from previous studies also revealed that the levels of total *COOLAIR,* and proximal and distal *COOLAIR* variants were reduced in the *cbfs* mutant in Col-0 or SW ecotype background when exposed to either short-term (3 and 24 hr) or long-term (14V) cold, which supports our results (*Figure 4—figure supplement 2*; *Park et al., 2018a*; *Song et al., 2021*). All these data support that *CBFs* are necessary to fully induce *COOLAIR* in the early phase of vernalization.

Consistent with the reduced *COOLAIR* levels in *cbfs* mutants, the *CBF3* overexpressor, *pSuper:CBF3-myc*, showed a much higher *COOLAIR* level than the wild type (Col-0), throughout the vernalization period (*Figure 4B*). It is noteworthy that *COOLAIR* expression in *pSuper:CBF3-myc* was further upregulated by cold, especially during the early vernalization phase (10V), and then suppressed during the late phase. This result implies that both the transcriptional and post-transcriptional regulations of CBFs are involved in the long-term cold response of *COOLAIR*.

A recent study showed that the first seasonal frost (< 0 °C) during the winter season strongly induces *COOLAIR* expression (*Zhao et al., 2021*). Therefore, we tested whether *CBFs* are also required for the strong induction of *COOLAIR* triggered by freezing temperatures. Wild-type and *cbfs* plants were subjected to NV, 10V, and 20V, with or without an additional 8 hr of −1 °C freezing treatment. Irrespective of the pre-treatment with non-freezing cold, the freezing temperature increased the levels of both *COOLAIR* and *CBF* in the wild type (*Figure 4C*, *Figure 4—figure supplement 3*). However, the *cbfs* mutants showed a much smaller increase in the *COOLAIR* level than the wild type when exposed to 8 hr of sub-zero cold after 10V and 20V (*Figure 4C*). In particular, *COOLAIR* levels were not elevated by freezing treatment in NV *cbfs*. Thus, *CBFs* seem responsible for both the gradual increase of *COOLAIR* during vernalization and the strong *COOLAIR* induction triggered by freezing temperatures.

## CRT/DREs at the 3′-end of *FLC* are necessary for CBFs-mediated induction of *COOLAIR* during vernalization

Since CBF3 could bind to the CRT/DREs in the first exon (DREc) and the 3′-end of *FLC* (DRE1 and 2) (*Figure 1B and C*), we investigated whether CRT/DRE is responsible for the CBF-mediated long-term cold response of *COOLAIR*. To address this, we performed an *A. thaliana* protoplast transfection assay using the *35S:CBF3-HA* effector construct and the *pCOOLAIR_{DRE}:LUC* reporter construct, in which the 1 kb wild-type *COOLAIR* promoter was fused to the luciferase reporter gene. The luciferase activity assay showed that CBF3-HA protein activated the transcription of the *COOLAIR* promoter (*Figure 5A*). In contrast, CBF3-HA failed to increase luciferase activity when the mutant version of the *COOLAIR* promoter with mutations in the DRE1 and 2 sequences (*pCOOLAIR_{DRE}^{m}:LUC*) was used as a reporter. Additionally, co-transfection of the *35S:CBF3-HA* effector and *pFLC:LUC* reporter construct, which harbors the 1 kb sequence with the promoter and first exon of *FLC*, did not show increased luciferase activity. These results strongly indicate that CBF3 can activate *COOLAIR* transcription through DRE1 and 2 located in the *COOLAIR* promoter, but not via the first exon of *FLC*.

To further determine whether the 3′-end sequence of *FLC* is required for cold-induction of *COOLAIR* expression, we measured *COOLAIR* expression before and after vernalization in the *FLC_{ΔCOOLAIR}* mutant (*FLC_{ΔCOOLAIR-1}*), that lacks a 324 bp portion of the *COOLAIR* promoter region (*Luo*

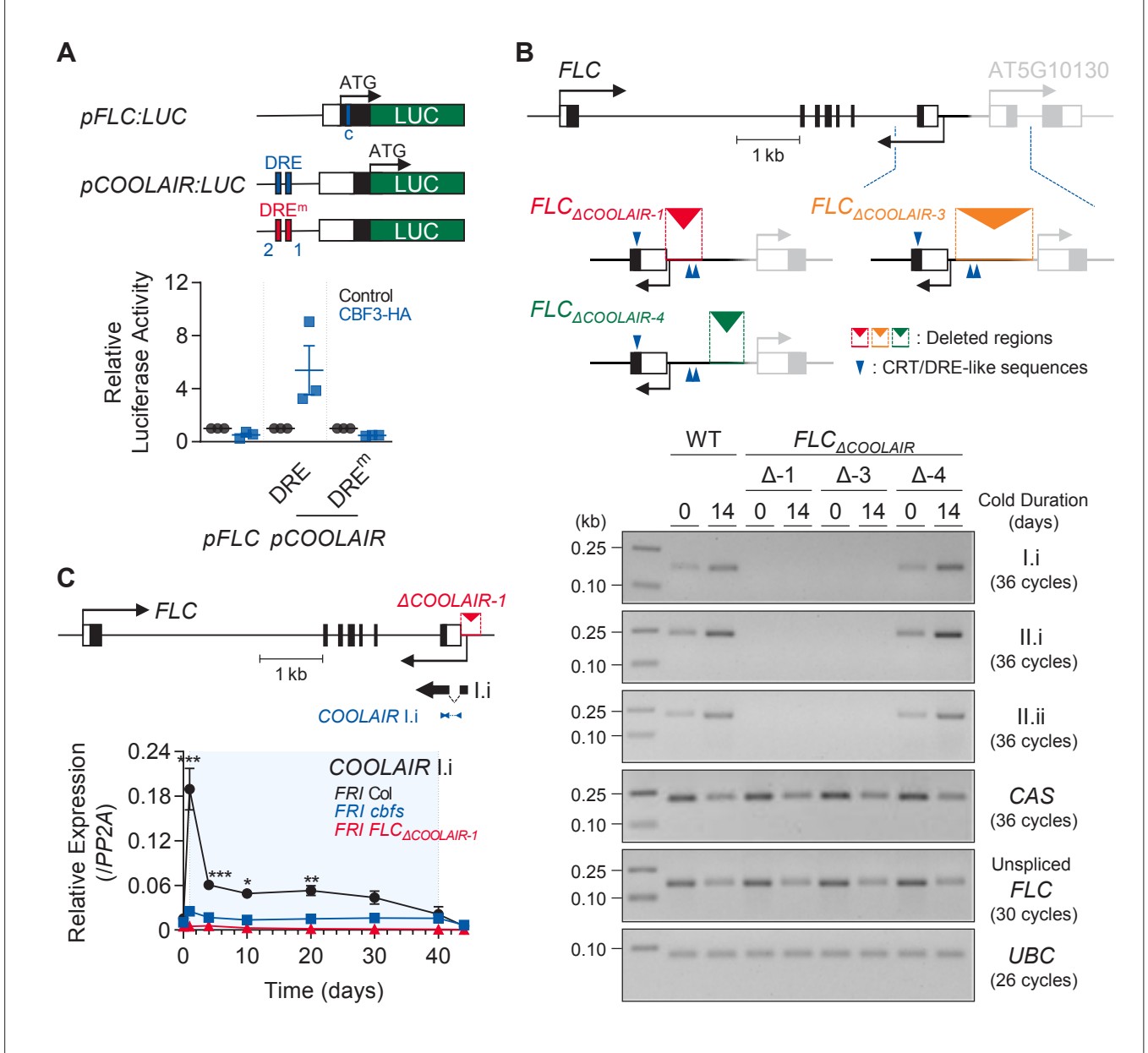

**Figure 5.** *COOLAIR* promoter region containing CRT/DREs is necessary for the *COOLAIR* induction by vernalization. (**A**) *Arabidopsis* protoplast transfection assay showing that CBF3 activates *COOLAIR* promoter with wild-type CRT/DRE (DRE) but fails to activate the one with mutant CRT/DRE (DRE^m). A schematic of the reporter constructs used for the luciferase assay is presented in the upper panel. *pFLC:LUC* contains 1 kb of the promoter, the 5'-UTR, and the first exon of *FLC*. The blue line in the *pFLC:LUC* graphic indicates the location of DREc. Wild-type and mutant forms of *pCOOLAIR:LUC* include 1 kb of the *COOLAIR* promoter with the 3'-UTR and the last exon of *FLC*. The blue and red lines mark the positions of the wild-type (DRE) and mutant (DRE^m) forms of DRE1 and DRE2, respectively. Each reporter construct was co-transfected into *Arabidopsis* protoplast together with the *35S:CBF3-HA* effector construct. In parallel, *35S-NOS* plasmid was transfected as a control. The result is shown below. Relative luciferase activities were normalized to that of the *35S-NOS* control. Values have been represented as mean ± SEM of three biological replicates. Dots and squares represent each data point. (**B**) Transcript levels of proximal (I.i) and distal (II.i and II.ii) *COOLAIR* isoforms in wild-type, *FLC_{ΔCOOLAIR-1}*, *FLC_{ΔCOOLAIR-3}*, and *FLC_{ΔCOOLAIR-4}* plants before and after 14 days of vernalization. *FLC_{ΔCOOLAIR-1}* and *FLC_{ΔCOOLAIR-3}* have a 324- and 685 bp deletion of the *COOLAIR* promoter region, respectively, where DRE1 and 2 are located. *FLC_{ΔCOOLAIR-4}* has a deletion in the 301 bp *COOLAIR* promoter region outside of DRE1 and 2 location. The positions of the deleted region are marked in red, orange, and green lines with reversed triangle in the upper graphic. Blue arrows denote CRT/DRE-like sequences. Black and gray bars denote exons, thin lines denote introns, and white bars denote UTRs of *FLC* and its neighboring gene (AT5G10130). Results of RT-PCR analysis are shown below. *UBC* was used as a quantitative control. (**C**) Transcript levels of proximal (I.i) *COOLAIR* isoform in wild-type, *cbfs-1*, and *FLC_{ΔCOOLAIR-1}* plants during vernalization. The position of the deleted region is marked in red lines with a reversed triangle in the upper graphic. Black bars denote exons, thin lines denote introns, and white bars denote UTRs of *FLC*. The thin black arrow below the gene structure indicates

*Figure 5 continued on next page*

*Figure 5 continued*

the transcriptional start site of *COOLAIR*. The thick black arrow below denotes exons of the type I.i *COOLAIR* variant. The position of the amplicon used for the qPCR analysis is marked with blue arrows. The result of qPCR analysis is presented in the lower panel. Relative transcript levels were normalized to that of *PP2A*. Values have been represented as mean ± SEM of three biological replicates. The blue shading indicates periods under cold treatment. Asterisks indicate a significant difference, as compared to NV (*, $p < 0.05$; **, $p < 0.01$; ***, $p < 0.001$; two-way ANOVA followed by Tukey's post-hoc test).

The online version of this article includes the following source data and figure supplement(s) for figure 5:

**Source data 1.** Uncropped labeled gel images and the original image files for the RT-PCR results.

**Figure supplement 1.** Genomic sequences near the deleted regions in two *COOLAIR* promoter deletion lines, $FLC_{\Delta COOLAIR-3}$ and $FLC_{\Delta COOLAIR-4}$.

**Figure supplement 2.** Levels of *CAS* and unspliced *FLC* are not significantly altered in the $FLC_{\Delta COOLAIR}$ mutant lines.

**Figure supplement 3.** Proximal *COOLAIR* level is not significantly increased by *CBF3* overexpression in the $FLC_{\Delta COOLAIR}$ mutant lacking CRT/DREs.

*et al., 2019*). Reverse transcription PCR (RT-PCR) analysis showed that neither proximal nor distal *COOLAIR* variants were detected in this mutant, and the levels did not increase after 14 day cold exposure (*Figure 5B*). Using the CRISPR-Cas9 system, we generated additional $FLC_{\Delta COOLAIR}$ mutant lines, $FLC_{\Delta COOLAIR-3,}$ and $FLC_{\Delta COOLAIR-4}$ (*Figure 5B*, *Figure 5—figure supplement 1*). $FLC_{\Delta COOLAIR-3}$ has a 685 bp deletion in the *COOLAIR* promoter region where both CRT/DREs are located. As expected, this mutant failed to express *COOLAIR* even after 14V, which is similar to the $FLC_{\Delta COOLAIR-1}$ mutant. In contrast, the $FLC_{\Delta COOLAIR-4}$ mutant, which bears a 301 bp deletion in the distal *COOLAIR* promoter region (upstream of CRT/DREs) and a 13 bp sequence replacement downstream of CRT/DREs, showed a normal vernalization-induced increase in all *COOLAIR* variants, similar to the wild type. This strongly suggests that CRT/DREs located at the *COOLAIR* promoter are critical for CBF-mediated *COOLAIR* induction during vernalization. Notably, the low-abundance convergent antisense transcripts (CAS; *Zhao et al., 2021*) were similarly reduced by 14V in all genotypes, that is, the wild type and $FLC_{\Delta COOLAIR}$ mutants (*Figure 5B*, *Figure 5—figure supplement 2*), suggesting their regulation is independent of *COOLAIR*.

Although the common *COOLAIR* TSS was eliminated in $FLC_{\Delta COOLAIR-1}$ (*Figure 5B*), the proximal *COOLAIR* isoform (type I.i) was still slightly detected by quantitative PCR (qPCR), probably due to alternative TSSs (*Figure 5C*). However, the expression of the proximal *COOLAIR* variant in the $FLC_{\Delta COOLAIR-1}$ mutant was not induced throughout the vernalization period, while the wild type showed type I.i *COOLAIR* peaks at 1V and 20V (*Figure 5C*). Furthermore, *CBF3* overexpression did not significantly increase type I.i *COOLAIR* levels in the $FLC_{\Delta COOLAIR-1}$ mutant, while it caused a strong increase in proximal *COOLAIR* in the wild-type background (Col-0) (*Figure 5—figure supplement 3*). Taken together, our results strongly suggest that the *COOLAIR* promoter region containing the two CRT/DREs is necessary for the long-term cold response of *COOLAIR*.

## CBFs-mediated *COOLAIR* induction during vernalization is not required for *FLC* silencing

Finally, we investigated the role of *COOLAIR* induced by CBFs in the vernalization-triggered *FLC* silencing process. *COOLAIR* was reported to facilitate the removal of H3K36me3 from *FLC* chromatin during vernalization, whereas the other lncRNAs, *COLDAIR* and *COLDWRAP*, are required for H3K27me3 deposition (*Csorba et al., 2014*; *Berry and Dean, 2015*; *Kim and Sung, 2017b*; *Zhu et al., 2021*). Therefore, we compared the enrichment of H3K4me3, H3K36me3, and H3K27me3 on *FLC* chromatin in non-vernalized or vernalized wild-type and *cbfs* plants. As reported (*Bastow et al., 2004*; *Sung and Amasino, 2004b*; *Yang et al., 2014*), the enrichments of H3K4me3 and H3K36me3 were reduced, but that of H3K27me3 was increased in *FLC* chromatin at 40V in the wild type (*Figure 6A–C*). However, although *COOLAIR* levels were significantly reduced in the *cbfs* mutant (*Figure 4A*), vernalization-mediated reductions in H3K4me3 and H3K36me3 levels and an increase in H3K27me3 levels in the *cbfs* mutant were comparable to those in the wild type (*Figure 6A–C*). This result suggests that CBF-mediated *COOLAIR* induction is not required for vernalization-induced epigenetic changes on *FLC* chromatin.

Consistent with the epigenetic changes, the *FLC* level in the *cbfs* mutant was gradually suppressed during the vernalization process, and the suppression was maintained after 40VT4 similar to that observed in the wild type (*Figure 6D*). Such result is consistent with the RNA-seq data previously reported using *cbfs* mutants in an SW ecotype (*Park et al., 2018a*). RNA-seq data analysis showed

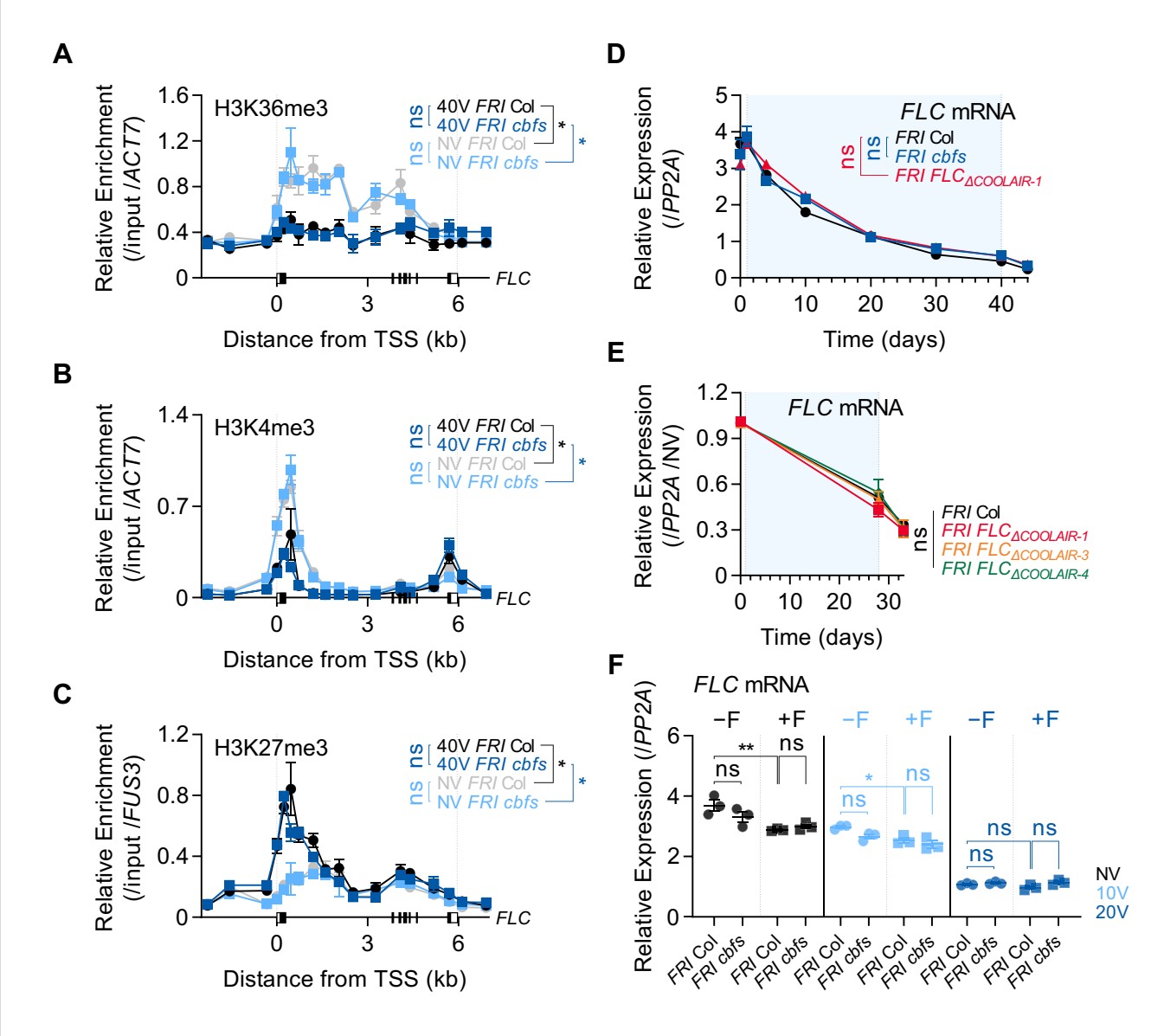

**Figure 6.** Vernalization-triggered epigenetic silencing of *FLC* is not affected by CBFs-mediated *COOLAIR* induction. (A–C) Enrichments of H3K36me3 (**A**), H3K4me3 (**B**), and H3K27me3 (**C**) on the *FLC* locus in NV or 40 V wild-type and *cbfs-1* plants. The whole seedlings were collected at ZT4 under an SD cycle. Modified histones were immunoprecipitated with anti-H3K36me3, anti-H3K4me3, or anti-H3K27me3 antibodies. H3K36me3, H3K4me3, and H3K27me3 enrichments of the IP/5% input were normalized to those of *ACT7* or *FUS3*. Relative enrichments have been represented as mean ± SEM of three biological replicates. One-way ANOVA and Tukey's post-hoc test were performed on the ChIP results obtained using two primer pairs corresponding to the *FLC* nucleation region (*Yang et al., 2017*). Asterisks indicate a significant difference (*, p < 0.05). ns, not significant ($P \geq 0.05$). (**D**) Transcript levels of *FLC* mRNA in wild-type, *cbfs-1* and *FLC*$_{\Delta COOLAIR-1}$ plants during and after vernalization. Blue shading denotes cold periods. Relative levels were normalized to that of *PP2A*. Values have been represented as mean ± SEM of three biological replicates. ns, not significant between wild type and mutants, as assessed using two-way ANOVA followed by Tukey's post-hoc test (p ≥ 0.05). (**E**) Transcript levels of *FLC* mRNA in wild type, *FLC*$_{\Delta COOLAIR-1}$, *FLC*$_{\Delta COOLAIR-3}$, and *FLC*$_{\Delta COOLAIR-4}$ during and after vernalization. Blue shading denotes cold periods. Relative levels were normalized to that of *PP2A*, and then normalized to NV of each genotype. Values have been represented as mean ± SEM of three biological replicates. ns, not significant between wild type and mutants, as assessed using two-way ANOVA followed by Tukey's post-hoc test (p ≥ 0.05). (**F**) Effect of the first frost-mimicking treatment (8 hr freezing) on the *FLC* mRNA level in wild type and *cbfs-1* after vernalization. NV, 10V, 20V, NV +F, 10V+F, and 20V+F plants were subjected to the treatments described in *Figure 4C*. Relative levels were normalized to that of *PP2A*. Values have been represented as mean ± SEM of three biological replicates. Dots and squares represent each data point. Asterisks indicate a significant difference (*, p < 0.05; **, p < 0.01; two-way ANOVA followed by Tukey's post-hoc test). ns, not significant (p ≥ 0.05).

that the *cbfs* mutant in the SW background exhibited reduced *FLC* levels after 14V, similar to the wild type (*Figure 4—figure supplement 2*).

Because our results indicate that *COOLAIR* induction is not required for *FLC* silencing and the function of *COOLAIR* in vernalization-induced *FLC* silencing is still debated (*Helliwell et al., 2011*; *Csorba et al., 2014*; *Zhao et al., 2021*; *Zhu et al., 2021*), we analyzed whether the loss-of-function mutations *FLC$_{\Delta COOLAIR-1}$* and *FLC$_{\Delta COOLAIR-3}$* result in any defect in the vernalization response. In contrast to the transgenic lines used in previous reports, *FLC$_{\Delta COOLAIR-1}$* and *FLC$_{\Delta COOLAIR-3}$*, which have small deletions in the *COOLAIR* promoter region, are unlikely to cause any unexpected effects on *FLC* chromatin. *FLC$_{\Delta COOLAIR-4}$*, which has a deletion outside of the CRT/DRE region, exhibited a gradual decrease and eventual silencing of *FLC* by long-term cold, similar to the wild type (*Figure 6E*). Interestingly, both *FLC$_{\Delta COOLAIR-1}$* and *FLC$_{\Delta COOLAIR-3}$* mutants exhibited similar decreases and silencing of *FLC* by vernalization (*Figures 5B, 6D and E*). These results show that *FLC* is normally silenced by vernalization regardless of *COOLAIR* induction by cold.

Consistent with this, the *cbfs* mutant, compared to the wild type, did not show any difference in the reduction of *FLC* levels upon exposure to a freezing cold for 8 hr after NV, 10V, and 20V (*Figure 6F*).

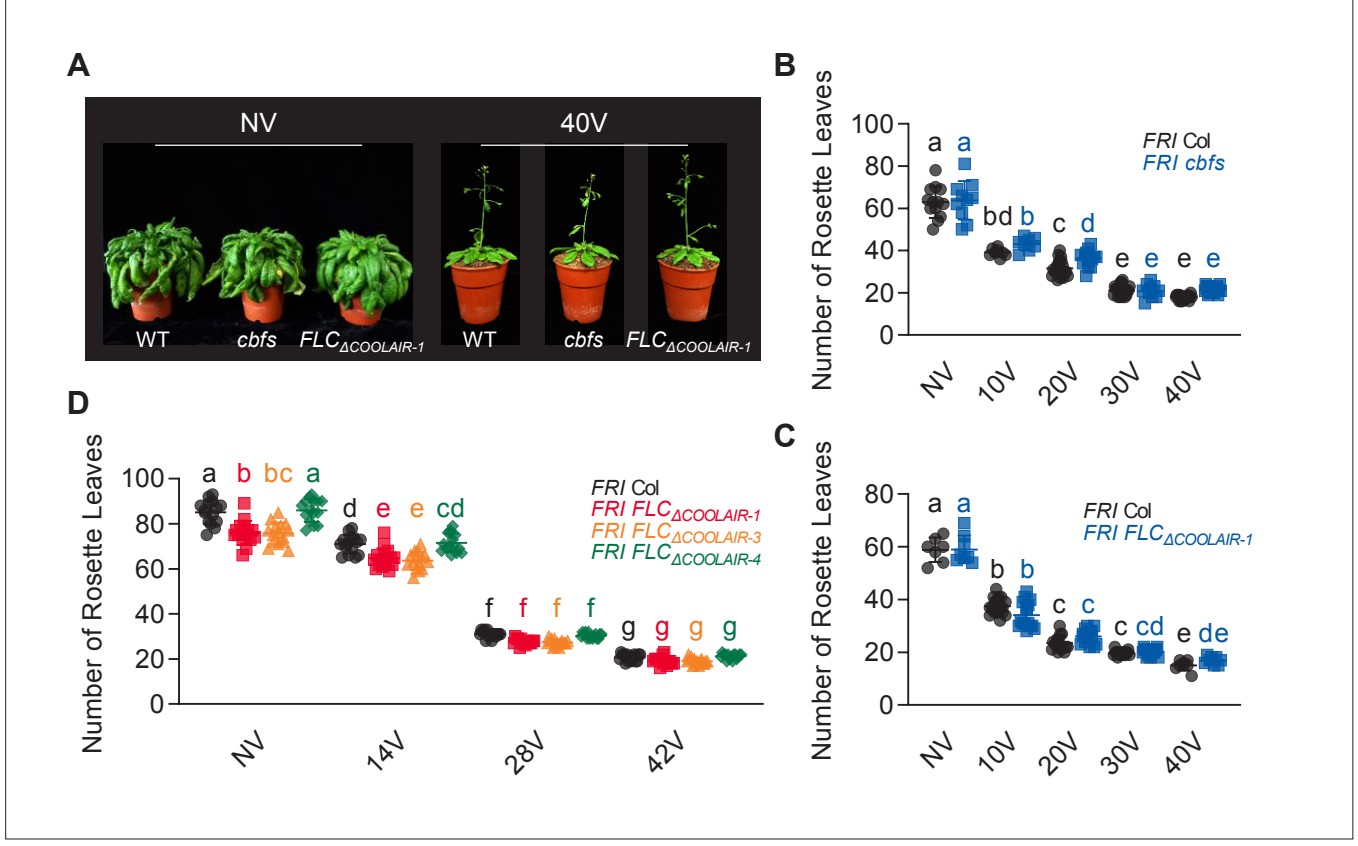

**Figure 7.** *CBF*-mediated *COOLAIR* induction under long-term cold is not absolutely necessary for vernalization response. (**A**) Photographs of NV and 40 V wild type, *cbfs-1*, and *FLC$_{\Delta COOLAIR-1}$*. NV plants were grown at 22 °C under LD and 40V plants were grown at 22 °C under LD after being treated for 40V under SD. The photos were taken when the NV plants started to bolt and the 40V plants showed the first open flower. (**B**) Flowering times of NV and vernalized (10V–40V) wild-type and *cbfs-1* plants. Flowering time was measured in terms of the number of rosette leaves produced when bolting. Plants were grown under LD after vernalization treatment. Bars and error bars represent the mean ± standard deviation (SD) of three biological replicates. Each dot and square indicates individual flowering time. Significant differences have been marked using different letters (a–e; two-way ANOVA followed by Tukey's post-hoc test; p < 0.05). (**C**) Flowering times of non-vernalized (NV) and vernalized (10V–40V) wild-type and *FLC$_{\Delta COOLAIR-1}$* plants. Flowering time was measured in terms of the number of rosette leaves produced when bolting. Plants were grown under LD after vernalization. Bars and error bars indicate the mean ± SD. Each dot and square represents the individual flowering time. Significant differences have been indicated using different letters (a–e; two-way ANOVA followed by Tukey's post-hoc test; P < 0.05). (**D**) Flowering times of non-vernalized (NV) and vernalized (14V–42V) wild-type, *FLC$_{\Delta COOLAIR-1}$*, *FLC$_{\Delta COOLAIR-3}$*, and *FLC$_{\Delta COOLAIR-4}$* plants. Flowering time was measured in terms of the number of rosette leaves produced when bolting. Plants were grown under LD after vernalization. Bars and error bars indicate the mean ± SD. Each dot, square, triangle, and polygon represents the individual flowering time. Significant differences have been indicated using different letters (a–g; two-way ANOVA followed by Tukey's post-hoc test; p < 0.05).

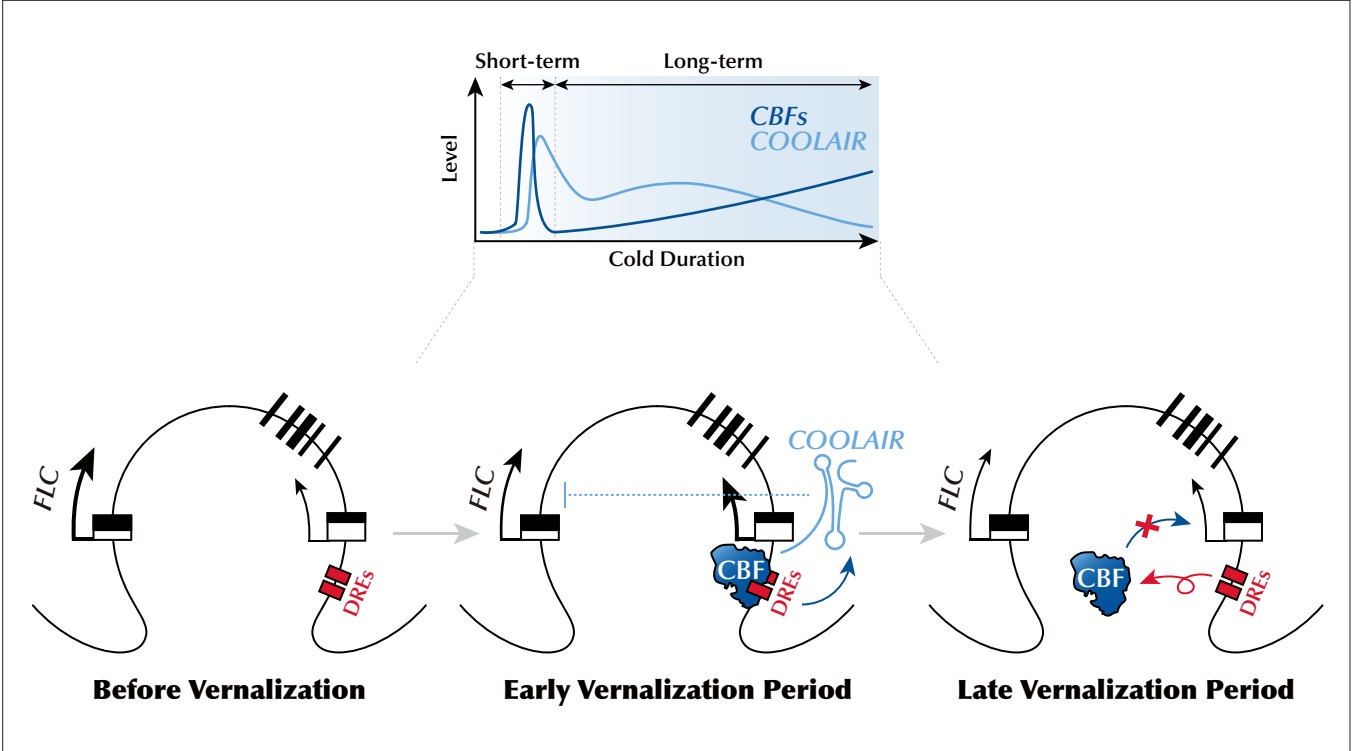

**Figure 8.** Schematic model describing the mechanism of CBFs-mediated *COOLAIR* expression during the early phase of vernalization. During the early phase of vernalization, increased CBF upregulates *COOLAIR* expression by binding to CRT/DREs at the 3'-end of *FLC*. In the late phase of vernalization, owing to the silencing of *FLC* chromatin, CBF proteins are released from the CRT/DREs in the *COOLAIR* promoter, which leads to a reduction in *COOLAIR* levels.

However, the same freezing cold treatment caused the failure of strong induction of *COOLAIR* in the *cbfs* (*Figure 4C*). These results also support that CBF-mediated *COOLAIR* induction is not required for *FLC* silencing by vernalization.

As expected from the *FLC* levels, vernalization-mediated promotion of flowering was comparable in the wild type and *cbfs* (*Figure 7A and B*), indicating that *cbfs* exhibits a normal vernalization response. Consistently, the flowering time of *FLC*ΔCOOLAIR-1 and *FLC*ΔCOOLAIR-3 was accelerated by vernalization, similar to that of the wild type (*Figure 7A, C and D*). These results show that *FLC* silencing occurs regardless of whether *COOLAIR* is induced during a cold exposure that results in vernalization. In summary, our data strongly support that CBFs, upregulated by long-term cold exposure, induce *COOLAIR* expression (*Figure 8*); however, cold-mediated *COOLAIR* induction is not required for *FLC* silencing during vernalization.

## Discussion

Winter is challenging for sessile plants: cold weather makes reproduction difficult, and frost threatens survival. Cold acclimation and vernalization, the two types of low-temperature memories, allow plants to overcome this harsh season by making them more viable during winter or flowering in the subsequent warm spring (*Thomashow, 1999*; *Michaels and Amasino, 2000*). Although it is well understood how plants recognize and remember short-term cold exposure to increase frost hardiness, there is still much to know about the long-term cold sensing mechanism that establishes the ability to flower.

*FLC* orthologs from a range of *A. thaliana* relatives express the vernalization-induced antisense lncRNA, *COOLAIR* (*Castaings et al., 2014*). As the proximal sequence blocks within the *COOLAIR* promoters are conserved, they are likely targets of cold sensor modules. This study showed that CBF proteins act as cold sensors binding to the CRT/DREs in the conserved promoter region, thereby activating the transcription of *COOLAIR*. Based on the data presented here, we propose a working model (*Figure 8*) in which a prolonged low-temperature environment gradually increases the levels of

CBF proteins, which activate *COOLAIR* transcription in the early phase of vernalization by binding to CRT/DREs at the 3'-end of *FLC*. However, as cold period extended, CBF proteins are excluded from the *COOLAIR* promoter, probably because silencing of *FLC* chromatin occurs regardless of CBFs level. This may account for a decrease in *COOLAIR* levels during the later phase of vernalization. Consistent with this, a previous report showed a continuous increase in the *COOLAIR* level throughout the vernalization period in the *vin3* mutant, which has a defect in the maintenance of *FLC* silencing (*Swiezewski et al., 2009*). Thus, VIN3-dependent Polycomb silencing is at least partially involved in the release of CBF proteins from the *COOLAIR* promoter.

Although CBFs are upregulated by cold to trigger *COOLAIR* expression, CBF-mediated *COOLAIR* induction during vernalization is not solely due to increased CBF levels because the *CBF3* overexpression lines still showed *COOLAIR* induction by vernalization (*Figure 4B*). This may indicate that additional factors participate in the regulation of CBFs function. As aforementioned, it has previously been reported that low temperatures cause functional activation of CBFs through the degradation of HD2C which maintains an epigenetically inactive state of downstream targets at warm temperatures (*Park et al., 2018b*). Moreover, cold triggers the monomerization or stabilization of CBF proteins, thereby promoting their functions (*Ding et al., 2018*; *Lee et al., 2021*). Similarly, such an additional regulatory mechanism of CBF function may be enhanced by long-term winter cold to fine-tune *COOLAIR* expression.

In addition, other thermosensors are likely to facilitate *COOLAIR* induction by vernalization. Recently, it was suggested that NTL8, an NAC domain-containing transcription factor, acts as an upstream regulator of *COOLAIR* (*Zhao et al., 2021*). NTL8 also binds to the *COOLAIR* promoter region, which is not far from CBF-binding sites (*O'Malley et al., 2016*; *Zhao et al., 2021*). *COOLAIR* is highly expressed in both dominant mutant alleles, *ntl8-Ds*, and *NTL8*-overexpressing transgenic plants at warm temperatures. Thus, it has been proposed that slow accumulation of the NTL8 protein, due to reduced dilution during slower growth at low temperatures, causes *COOLAIR* induction during vernalization. In addition, a WRKY transcription factor, WRKY63, has been suggested as another inducer of *COOLAIR* transcription during vernalization (*Hung et al., 2022*). WRKY63 is enriched in the *COOLAIR* promoter region, and *COOLAIR* expression is reduced in the *wrky63* mutant, suggesting that WRKY63 functions as a *COOLAIR* activator. Unlike NTL8, both transcript and protein levels of *WRKY63* were upregulated by vernalization, which may lead to an increase in *COOLAIR* expression. Thus, it would be worthwhile to elucidate whether NTL8, WRKY63, and CBFs synergistically activate *COOLAIR* transcription during vernalization. The reason why vernalization-triggered *COOLAIR* induction requires more than one thermosensor is unclear. Presumably, multiple thermosensory modules ensure precise and robust *COOLAIR* expression during winter, when the temperature fluctuates.

Transcription of *COOLAIR* has been proposed to govern the *FLC* chromatin environment in both warm and cold temperatures (*Csorba et al., 2014*; *Fang et al., 2020*; *Xu et al., 2021b*). Previous studies have inferred that *COOLAIR* is required to remove H3K36me3 from *FLC* chromatin, especially under low-temperature conditions. This argument is supported by the delayed H3K36me3 removal in vernalized *FLC-TEX* transgenic plants, where the *COOLAIR* promoter is substituted by the *rbcs3B* terminator sequence (*Csorba et al., 2014*; *Marquardt et al., 2014*; *Wang et al., 2014*). In addition, *COOLAIR* is reported to promote condensation of FRI upon cold exposure (*Zhu et al., 2021*). Thus, FRI is sequestered from the *FLC* promoter as a nuclear condensate, which causes the transcriptional repression of *FLC*. Therefore, it has been suggested that vernalization-induced *COOLAIR* prevents the activation of *FLC* by the FRI complex and causes epigenetic silencing of *FLC* chromatin through a decrease of H3K36me3 during the vernalization process.

However, there is some controversy as to whether *COOLAIR* is required for vernalization and the associated *FLC* silencing. In contrast to the *FLC-TEX* lines, some T-DNA insertion lines which lack *COOLAIR* induction by long-term cold exposure showed a normal vernalization response (*Helliwell et al., 2011*). Likewise, the results from this study show that *COOLAIR* transcription is not required for the epigenetic silencing of *FLC*. The *cbfs* mutant, which exhibits severely reduced *COOLAIR* induction, shows relatively normal vernalization responses such that flowering time is fairly well accelerated and active histone marks, H3K36me3 and H3K4me3, are reduced in *FLC* chromatin while the repressive histone mark, H3K27me3, is increased in *FLC* chromatin similar to the wild type (*Figures 6 and 7*). Consistently, *FLC$_{\Delta COOLAIR-1}$*, and *FLC$_{\Delta COOLAIR-3}$* mutants, which have almost undetectable levels of *COOLAIR*, exhibit both *FLC* suppression and flowering acceleration by vernalization similar to that

seen in the wild type (*Figures 6 and 7*). Moreover, the freezing treatment mimicking the first frost did not affect *FLC* suppression in the *cbfs*, although the mutant showed defects in CBF-mediated *COOLAIR* induction (*Figures 4C and 6F*). It is noteworthy that our finding that vernalization and the associated *FLC* silencing do not require *COOLAIR* expression was obtained using *cbfs* and *FLC$_{\Delta COOLAIR}$* mutant lines rather than the transgenic lines used in previous studies. Thus, we can exclude the possibility that transgenes somehow provide cryptic promoters or cause unexpected epigenetic effects.

*COOLAIR* is highly conserved among *Arabidopsis* relatives (*Castaings et al., 2014*); thus, our finding that *COOLAIR* is not required for vernalization-induced *FLC* silencing is surprising. There are several caveats in the studies of *COOLAIR* function. First, different experimental schemes may affect the vernalization response differently. A recent study of plants grown in fluctuating temperatures designed to mimic natural environments may indicate a role for *COOLAIR* in vernalization-mediated *FLC* repression (*Zhao et al., 2021*), and perhaps fluctuating temperatures that mimic field conditions may have different effects on the vernalization response of *cbfs* or *FLC$_{\Delta COOLAIR}$* mutants as well. However, the experiments in most studies that claimed a role for *COOLAIR* in vernalization were performed in laboratory conditions with controlled temperatures as in this study. If indeed *COOLAIR* does have a role in vernalization in specific environmental conditions, the lack of a role for *COOLAIR* in vernalization in other conditions such as those in this study may be due to functional redundancy between *COOLAIR* and other *FLC* repressors such that other repressors, such as *COLDAIR*, obscure the effect of *COOLAIR* on *FLC* silencing, as has been proposed before (*Castaings et al., 2014*). This issue should be addressed in future studies.

Although short-term and long-term cold memories elicit different developmental responses, both are triggered by the same physical environment. Therefore, it is not surprising that a common signaling network governs both processes. *CBF* genes have been highlighted for their roles in cold acclimation (*Gilmour et al., 1998*; *Liu et al., 1998*; *Jia et al., 2016*; *Zhao et al., 2016*). CBFs rapidly accumulate upon short-term cold exposure, thereby activating the transcription of cold-responsive genes, such as *COR15A* and *RD29A* (*Stockinger et al., 1997*; *Liu et al., 1998*). Interestingly, CBFs are also highly accumulated during vernalization and mediate long-term cold-induced *COOLAIR* expression. Hence, our results indicate that the cold signaling pathways that establish the short-term and long-term cold responses are not sharply distinguishable, as suggested previously (*Sung and Amasino, 2004a*; *Seo et al., 2009*).

## Materials and methods
### Plant materials and treatments

*Arabidopsis thaliana* Columbia-0 (Col-0), Wassilewskija-2 (Ws-2), and Col *FRI$^{Sf2}$* (*FRI* Col) were used as the wild types in this study. "Wild type" in this paper mainly refers to *FRI* Col unless otherwise specified. The *FRI* Col (*Lee et al., 1994*), *cbfs-1*, *pCBF3:CBF3-myc* (*Jia et al., 2016*), *pSuper:CBF3-myc* (*Liu et al., 2017*), *FLC$_{\Delta COOLAIR-1}$* (*Luo et al., 2019*), *35S:CBF1*, *35S:CBF2*, and *35S:CBF3* *Gilmour et al., 2004* have been previously described. *FRI cbfs-1* was generated by crossing the *FRI* Col with *cbfs-1* stated above. *FRI FLC$_{\Delta COOLAIR-1}$* was generated by crossing *FRI* Col with *FLC$_{\Delta COOLAIR-1}$* in the Col-0 background, as described above. *pSuper:CBF3-myc FLC$_{\Delta COOLAIR-1}$* was generated by crossing *pSuper:CBF3-myc* with *FLC$_{\Delta COOLAIR-1}$*.

For all experiments using plant materials, seeds were sown on half-strength Murashige and Skoog (MS) medium (Duchefa, Haarlem, Netherlands) containing 1% sucrose and 1% plant agar (Duchefa) and stratified at 4 °C for 3 days. Plants were germinated and grown at 22 °C under a short-day (SD) cycle (8 hr light/16 hr dark) with cool white fluorescent lights (100 μmol·m$^{-2}$·s$^{-1}$). For the vernalization treatment, the plants were grown at 22 °C for the prescribed period and then transferred to a 4 °C growth chamber under an SD cycle. To adjust the developmental stage, the plants vernalized for 40 days (40V), 30 days (30V), 20 days (20V), 10 days (10V), 4 days (4V), and 1 day (1V) were transferred to 4 °C after being grown at 22 °C for 10, 12, 14, 15, 15, and 16 days, respectively. Non-vernalized (NV) plants were grown at 22 °C for 16 days. For 0.5, 1, 1.5, 2, 3, 6, and 24 hr of cold treatment, the plants were grown at 22 °C for 16 days before the 4 °C treatment. For the freezing treatment, NV, 10V, and 20V plants were transferred to −1 °C under dark condition for 8 hr (F). To measure flowering time of vernalized plants, plants were transferred to 22 °C under a long-day (LD) cycle (16 hr light/8 hr dark) directly after the vernalization treatment.

## Generation of $FLC_{\Delta COOLAIR}$ lines by CRISPR-Cas9

The construction of $FLC_{\Delta COOLAIR-3}$ and $FLC_{\Delta COOLAIR-4}$ are as previously described (*Luo et al., 2019*). Briefly, two pairs of sgRNAs were designed for the deletion of *COOLAIR* promoter regions (5′-CTTCACAGT GAAGAAGCCTA-3′ and 5′-AAATGCACTCTTACGTAACG-3′ for $FLC_{\Delta COOLAIR-3}$; 5′-TTATCCTAAACGC GTATGGT-3′ and 5′-CGTAGTTCCGTCATCCATGA-3′ for $FLC_{\Delta COOLAIR-4}$), and sgRNA-Cas9 cassettes were introduced into *Arabidopsis* by floral dipping. Homozygous deletion mutants were screened in $T_2$ generation, and sgRNA-Cas9 free mutants were further isolated in $T_3$ generation.

## Plasmid construction

To generate the *35S:CBF3-HA* construct, the entire coding sequence of CBF3 was amplified using PCR and cloned into the *HBT-HA-NOS* plasmid (*Yoo et al., 2007*). For $pCOOLAIR_{DRE}:LUC$ and $pCOOLAIR_{DRE}{}^m:LUC$ construction, the 1 kb sequence containing the *COOLAIR* promoter, the 3′-untranslated region (UTR), and the last exon of *FLC* was amplified by PCR and cloned into the *LUC-NOS* plasmid (*Hwang and Sheen, 2001*). To generate $pCOOLAIR_{DRE}{}^m:LUC$, two CRT/DREs in the *COOLAIR* promoter, 5′-AGGTCGGT-3′ and 5′-ACCGACAT-3′, were replaced with 5′-CGAGGTGT-3′ and 5′-TGAACCCA-3′, respectively. For *pFLC:LUC* construction, the 1 kb sequence containing the promoter, 5′-UTR, and the first exon of *FLC* was amplified by PCR and was cloned into the *LUC-NOS* plasmid.

## EMSA

Maltose-binding protein (MBP) and MBP-CBF3 recombinant fusion proteins were induced by 500 μM isopropyl β-D-thiogalactoside (IPTG) in the *Esecherichia coli* BL21 (DE3) strain. Cell extracts were isolated with a buffer containing 20 mM Tris-HCl (pH 7.4), 1 mM EDTA, 200 mM NaCl, 10% glycerol, 1 mM DTT, and 1 mM PMSF. Proteins were purified from cell extracts using MBPTrap HP column (GE Healthcare, Chicago, IL, USA) with ÄKTA FPLC system (Amersham Biosciences, Amersham, UK). The Cy5-labeled probes and unlabeled competitors were generated by annealing 25 bp-length oligonu-cleotides. Electrophoretic mobility shift assay (EMSA) was performed as previously described with a few modifications (*Seo et al., 2012*). For each EMSA reaction, 5 μM of protein and 100 nM of Cy5-labeled probe were incubated at room temperature in a binding buffer containing 10 mM Tris-HCl (pH 7.5), 1 mM EDTA, 50 mM NaCl, 5% glycerol, and 5 mM DTT. For the competition assay, the reaction mixtures were incubated in the presence of 100 fold molar excess of each competitor. The reaction mixtures were electrophoresed at 100 V after incubation. The Cy5 signals were detected using WSE-6200H LuminoGraph II (ATTO, Amherst, NY, USA).

## ChIP assay

Approximately 4 g of seedlings grown on MS plates was collected and cross-linked using 1% (v/v) formaldehyde. Nuclei were isolated from seedlings using a buffer containing 20 mM PIPES-KOH (pH 7.6), 1 M hexylene glycol, 10 mM $MgCl_2$, 0.1 mM EGTA, 15 mM NaCl, 60 mM KCl, 0.5% Triton X-100, 5 mM β-mercaptoethanol, and 1×cOmplete EDTA-free Protease Inhibitor Cocktail (Roche, Basel, Switzerland) as described previously with a few modifications (*Shu et al., 2014*). Isolated nuclei were lysed with a buffer containing 50 mM Tris-HCl (pH 7.4), 150 mM NaCl, 1% (v/v) Triton X-100, and 1% SDS (w/v), and were subsequently sonicated using a Digital Sonifier (Branson, Danbury, CT, USA). Sheared chromatin was diluted with a buffer containing 50 mM Tris-HCl (pH 7.4), 150 mM NaCl, 1% (v/v) Triton X-100, and 1 mM EDTA. Chromatin immunoprecipitation (ChIP) was performed by incubating sheared chromatin with Protein G Sepharose 4 Fast Flow beads (GE Healthcare) and anti-bodies. Anti-myc (MBL, Woburn, MA, USA; #M192-3) and normal mouse $IgG_1$ (Santa Cruz Biotech-nology, Santa Cruz, CA, USA; #sc-3877) were used to detect CBF3 protein enrichment at the *FLC* locus, and anti-H3K36me3 (Abcam, Cambridge, UK; #Ab9050), anti-H3K4me3 (Millipore, Bedford, MA, USA; #07–473), and anti-H3K27me3 (Millipore; #07–449) were used to detect histone modifica-tions on the *FLC* chromatin. DNA was extracted with phenol:chloroform:isoamyl alcohol (25:24:1, v/v) or Chelex 100 Resin according to the manufacturer's instruction.

## Gene expression analysis

Total RNA was isolated from –100 mg of seedlings grown on MS agar plate using TRI Reagent (Sigma-Aldrich, St Louis, MO, USA) or TS Plant Mini Kit (Taeshin Bio Science, Gyeonggi-do, Korea). cDNA was generated using 4 μg of total RNA, 5 units of recombinant DNase I (TaKaRa, Kyoto, Japan), 200 units

of RevertAid reverse transcriptase (Thermo Scientific, Waltham, MA, USA), and buffer containing 0.25 mM dNTP and 0.1 μg oligo(dT)$_{18}$. Quantitative PCR analysis was performed using iQ SYBR Green Supermix (Bio-Rad, Hercules, CA, USA) and a CFX96 Real-Time PCR system (Bio-Rad).

To examine *COOLAIR* expression in the *COOLAIR* promoter deletion lines, Total RNAs were extracted using the RNeasy Plus Mini Kit (Qiagen, Hilden, Germany) according to the manufacturer's instructions, followed by the digestion of residual genomic DNA by the gDNA wiper (Vazyme, Nanjing, China). 1.0 μg RNA from each sample was taken for cDNA synthesis using the HiScript III 1st Strand cDNA Synthesis Kit (Vazyme) with the transcript-specific primers (5'-TGGTTGTTATTTGGTGG TGTGAA-3' for *COOLAIR* class I and 5'- GCCCGACGAAGAAAAAGTAG-3' for class II; *Zhao et al., 2021*). Semi-quantitative PCR amplifications were performed, followed by agarose gel separation of PCR products.

## Immunoblotting

Total protein was isolated from –100 mg of seedlings using a buffer containing 50 mM Tris-HCl (pH 7.5), 100 mM NaCl, 10 mM MgCl$_2$, 1 mM EDTA, 10% glycerol, 1 mM PMSF, 1 mM DTT and 1×cOmplete EDTA-free Protease Inhibitor Cocktail (Roche). Fifty micrograms of total protein were loaded onto SDS-PAGE gels and separated by electrophoresis. The proteins were transferred to PVDF membranes (Amersham Biosciences) and probed with an anti-myc (MBL; #M192-3; 1:10,000 dilution) antibody overnight at 4 °C. The samples were then probed with horseradish peroxidase (HRP)-conjugated anti-mouse IgG (Cell Signaling, Danvers, MA, USA; #7076; 1:10,000 dilution) antibody at room temperature. The signals were detected with ImageQuant LAS 4000 (GE Healthcare) using WesternBright Sirius ECL solution (Advansta, San Jose, CA, USA).

## RNA-seq data analysis

We retrieved two sets of RNA sequencing (RNA-seq) data deposited in the National Center for Biotechnology Information (NCBI) under BioProject accession codes PRJNA416120 and PRJNA732005. PRJNA416120 contains raw reads from NV, 1V, and 14V wild-type and *cbfs* plants in a Sweden-ecotype (SW) background (*Park et al., 2018a*). PRJNA732005 contains reads from 0, 3, and 24 hr cold-treated wild-type and *cbfs* plants in a Col-0 background (*Song et al., 2021*). The reads were aligned to the *Arabidopsis* TAIR 10 reference genome and annotated in ARAPORT 11 using STAR version 2.7.10 a. Isoform estimation was performed using Salmon version 1.6.0.

## Luciferase assay using *Arabidopsis* protoplast

Protoplast isolation and transfection were performed as previously described with some modifications (*Yoo et al., 2007*). Protoplasts were isolated from leaves of SD-grown wild-type plant (Col-0) using a buffer containing 150 mg Cellulase Onozuka R-10 (Yakult, Tokyo, Japan), 50 mg Maceroenzyme R-10 (Yakult), 20 mM KCl, 20 mM MES-KOH (pH 5.6), 0.4 M D-mannitol, 10 mM CaCl$_2$, 5 mM β-mercaptoethanol, and 0.1 g bovine serum albumin. For protoplast transfection, 200 μg of plasmid DNA and isolated protoplasts were incubated in a buffer containing 0.1 M D-mannitol, 50 mM CaCl$_2$, and 20% (w/v) PEG. Luciferase activity in the protoplasts was measured using the Luciferase Assay System (Promega, Madison, WI, USA) and MicroLumat Plus LB96V microplate luminometer (Berthold Technologies, Bad Wildbad, Germany).

## Acknowledgements

We thank Dr. Shuhua Yang (China Agricultural University, China) for providing the *cbfs-1*, *pCBF3:CBF3-myc*, and *pSuper:CBF3-myc* seeds. This work was supported by the Cooperative Research Program for Agriculture Science and Technology Development (No. PJ01315201 and PJ01315401), Rural Development Administration, Republic of Korea. This work was also supported by National Research Foundation of Korea (NRF) grants funded by the Korean government (MSIT) (No. 2019R1A2C2004313, 2021R1A5A1032428, and 2022R1A2C1091491). Y He's laboratory was funded by the grants from the National Natural Science Foundation of China (No. 31830049 and 31721001). M Jeon and D Jeong were supported by Brain Korea 21 Plus Project. We thank Editage (http://www. editage.co.kr) for English language editing.

## Additional information

### Funding

| Funder | Grant reference number | Author |
|---|---|---|
| National Research Foundation of Korea | 2019R1A2C2004313 | Ilha Lee |
| National Research Foundation of Korea | 2022R1A2C1091491 | Ilha Lee |
| National Natural Science Foundation of China | 31830049 | Yuehui He |
| National Natural Science Foundation of China | 31721001 | Yuehui He |
| National Research Foundation of Korea | 2021R1A5A1032428 | Ilha Lee |

The funders had no role in study design, data collection and interpretation, or the decision to submit the work for publication.

### Author contributions

Myeongjune Jeon, Conceptualization, Data curation, Formal analysis, Validation, Investigation, Visualization, Methodology, Writing - original draft, Writing – review and editing; Goowon Jeong, Conceptualization, Data curation, Formal analysis, Validation, Investigation, Methodology, Writing – review and editing; Yupeng Yang, Xiao Luo, Conceptualization, Data curation, Formal analysis, Validation, Investigation, Visualization, Methodology; Daesong Jeong, Validation, Investigation, Visualization, Methodology; Jinseul Kyung, Youbong Hyun, Validation, Writing – review and editing; Yuehui He, Ilha Lee, Conceptualization, Resources, Data curation, Supervision, Funding acquisition, Validation, Writing - original draft, Project administration, Writing – review and editing

### Author ORCIDs

Myeongjune Jeon ⓘ http://orcid.org/0000-0001-9195-1810
Goowon Jeong ⓘ http://orcid.org/0000-0002-5023-920X
Daesong Jeong ⓘ http://orcid.org/0000-0003-0224-1687
Jinseul Kyung ⓘ http://orcid.org/0000-0002-8305-704X
Ilha Lee ⓘ http://orcid.org/0000-0002-3516-4326

### Decision letter and Author response

Decision letter https://doi.org/10.7554/eLife.84594.sa1
Author response https://doi.org/10.7554/eLife.84594.sa2

## Additional files

### Supplementary files

- Supplementary file 1. Oligonucleotide sequences used in this study.
- MDAR checklist

### Data availability

No new data have been generated for this manuscript. Previously published datasets used for this study are deposited in NCBI, under BioProject accession codes PRJNA416120 and PRJNA732005.

The following previously published datasets were used:

| Author(s) | Year | Dataset title | Dataset URL | Database and Identifier |
|---|---|---|---|---|
| Park S, Gilmour SJ, Grumet R, Thomashow MF | 2018 | Potential role of the CBF Pathway contributing to local adaptation of ecotypes collected from Italy and Sweden (thale cress) | https://www.ncbi.nlm.nih.gov/bioproject/PRJNA416120 | NCBI BioProject, PRJNA416120 |
| Song Y, Zhang X, Li M, Yang H, Fu D, Lv J, Ding Y, Gong Z, Shi Y, Yang S | 2021 | Genome-wide identification of CBFs targets in Arabidopsis | https://www.ncbi.nlm.nih.gov/bioproject/PRJNA732005/ | NCBI BioProject, PRJNA732005 |

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
