## [Editor Report]

This important work advances our understanding of the systems that plants evolved to coordinate developmental processes such as the timing of flowering with seasonal change, particularly with respect to the regulation and role of long non-coding RNAs (lncRNAs) complementary to genes encoding proteins that regulate developmental switches. The evidence supporting the conclusions is solid. The work will be of interest to those interested plant development as well those interested in the role and regulation of lncRNAs.

---

## [Decision Letter]

**Decision letter after peer review:**

Thank you for submitting your article "Vernalization-triggered expression of the antisense transcript COOLAIR is mediated by CBF genes" for consideration by *eLife*. Your article has been reviewed by 3 peer reviewers, including Richard Amasino as Reviewing Editor and Reviewer #1, and the evaluation has been overseen by Jürgen Kleine-Vehn as the Senior Editor. The following individuals involved in the review of your submission have agreed to reveal their identity: Iain Searle (Reviewer #2); Chris A. Helliwell (Reviewer #3).

Essential revisions:

I would like you to respond to the following points of one of the reviewers.

1. It wasn't clear why the pCBF3:CBF3-myc lines were not used for ChIP assays to look at the binding of CBF3 to the COOLAIR promoter, this would allow a more realistic assessment of CBF3 interaction with the COOLAIR promoter under native conditions.

2. Figure 4. Why isn't the same scale used for the COOLAIR expression levels in parts A and B of this Figure? This would allow more direct comparison between the CBF3 over-expressors, wild type, and cbf1 mutants.

3. Figure 5. Do the authors have a qPCR assay for the various COOLAIR transcripts rather than the semi-quantitative gel-based assay used in this figure?

4. Figure 5. The conclusions of the deletion analysis would be stronger if specific deletions of the actual CRT/DRE elements were made, and this is likely to be possible with gene editing techniques.

5. I wasn't sure that Figure 8 really added anything to the paper as there isn't any new information on the actual function of COOLAIR.

*Reviewer #1 (Recommendations for the authors):*

Nothing to add. The work seems solid. I understand it has been submitted before so in this version the authors have addressed previous concerns.

*Reviewer #2 (Recommendations for the authors):*

There is very compelling data presented to support the manuscript's conclusions. Unusually, I have no suggestions for further experiments to support the paper's conclusions.

I found the manuscript very clearly written, the data elegantly presented, and no typographical errors.

The authors should be commended on the quality of the presented experimental data, and for challenging a widely accepted model by the research community.

*Reviewer #3 (Recommendations for the authors):*

1. It wasn't clear why the pCBF3:CBF3-myc lines were not used for ChIP assays to look at the binding of CBF3 to the COOLAIR promoter, this would allow a more realistic assessment of CBF3 interaction with the COOLAIR promoter under native conditions.

2. Figure 4. Why isn't the same scale used for the COOLAIR expression levels in parts A and B of this Figure? This would allow more direct comparison between the CBF3 over-expressors, wild type, and cbf1 mutants.

3. Figure 5. Do the authors have a qPCR assay for the various COOLAIR transcripts rather than the semi-quantitative gel-based assay used in this figure?

4. Figure 5. The conclusions of the deletion analysis would be stronger if specific deletions of the actual CRT/DRE elements were made, this is likely possible with gene editing techniques.

5. I wasn't sure that Figure 8 really added anything to the paper as there isn't any new information on the actual function of COOLAIR.

6. Given the differences in responses observed between this study and other it would have been informative if experiments (particularly to measure FLC and flowering time) had been carried out in different growth conditions such as in common gardens or cabinets programmed to mimic natural conditions. This may have been helpful in identifying conditions where COOLAIR transcription is required or reinforced the conclusion that it is not an essential component of the vernalization response.

---

## [Author Response]

Essential revisions:I would like you to respond to the following points of one of the reviewers.1. It wasn't clear why the pCBF3:CBF3-myc lines were not used for ChIP assays to look at the binding of CBF3 to the COOLAIR promoter, this would allow a more realistic assessment of CBF3 interaction with the COOLAIR promoter under native conditions.

We aimed to show the binding affinity of CBF3 protein to the *COOLAIR* promoter during vernalization. The cold-triggered changes in *CBF3* transcript level in *pCBF3:CBF3-myc* would make us difficult to simply address this. We, therefore, thought the *pSuper:CBF3-myc* line is more proper than *pCBF3:CBF3-myc* for ChIP assays to check the CBF3 protein binding to the *COOLAIR* promoter regardless of the changes in *CBF3* transcript levels. To make this point clear, we have added this statement into the Result section (line no. 164-167).

2. Figure 4. Why isn't the same scale used for the COOLAIR expression levels in parts A and B of this Figure? This would allow more direct comparison between the CBF3 over-expressors, wild type, and cbf1 mutants.

The *cbfs* mutant is in the *FRI* Col background, while the *pSuper:CBF3-myc* line is in Col. The basal *COOLAIR* level is known to be affected by the presence/absence of a functional *FRI* allele (Swiezewski et al., 2009). Thus, it is not directly comparable for the *COOLAIR* levels among *cbfs,* wild type (Col and *FRI*-Col), and *pSuper:CBF3-myc*.

3. Figure 5. Do the authors have a qPCR assay for the various COOLAIR transcripts rather than the semi-quantitative gel-based assay used in this figure?

We performed RT-qPCR assays to assess *COOLAIR* variant levels in *FLC_ΔCOOLAIR_* lines. However, as the *COOLAIR* levels in the *FLC_ΔCOOLAIR-1_* and *FLC_ΔCOOLAIR-3_* lines were too low to reliably detect by RT-qPCR and did not increase by 14-d cold treatment, we presented the data as DNA gel images. Instead, we have added the RT-qPCR results for *CAS* and unspliced *FLC* levels in *FLC_ΔCOOLAIR_* lines as shown in Figure 5—figure supplement 2.

4. Figure 5. The conclusions of the deletion analysis would be stronger if specific deletions of the actual CRT/DRE elements were made, and this is likely to be possible with gene editing techniques.

We agree that the specific mutations of CRT/DREs would provide stronger evidence. However, unfortunately we were unable to generate the mutant and test its vernalization response on time. Instead, the results of luciferase assay using the reporter with specific mutations in CRT/DREs shown in Figure 5A may partially address the concern.

5. I wasn't sure that Figure 8 really added anything to the paper as there isn't any new information on the actual function of COOLAIR.

We agree that Figure 8 does not have any new information. But we believe the model in Figure 8 summarizing our results would allow readers to catch the take home message more easily and clearly.

Reviewer #1 (Recommendations for the authors):Nothing to add. The work seems solid. I understand it has been submitted before so in this version the authors have addressed previous concerns.

We appreciate for the positive assessment.

Reviewer #2 (Recommendations for the authors):There is very compelling data presented to support the manuscript's conclusions. Unusually, I have no suggestions for further experiments to support the paper's conclusions.I found the manuscript very clearly written, the data elegantly presented, and no typographical errors.The authors should be commended on the quality of the presented experimental data, and for challenging a widely accepted model by the research community.

Thank you for the positive evaluations and encouragement.

Reviewer #3 (Recommendations for the authors):1. It wasn't clear why the pCBF3:CBF3-myc lines were not used for ChIP assays to look at the binding of CBF3 to the COOLAIR promoter, this would allow a more realistic assessment of CBF3 interaction with the COOLAIR promoter under native conditions.

We deeply agree with your concern that using endogenous promoter would provide a more realistic assessment of CBF3 interaction. However, what we actually aimed to show in Figure 1C was a change in the binding affinity of CBF3 protein to the *COOLAIR* promoter throughout the vernalization period. The vernalization-induced increase of *CBF3* transcript level in *pCBF3:CBF3-myc* would make us difficult to simply address this. We, therefore, thought the *pSuper:CBF3-myc* line is more proper than *pCBF3:CBF3-myc* for ChIP assays. To make this point clear, we have added this statement into the Result section (line no. 164-167).

2. Figure 4. Why isn't the same scale used for the COOLAIR expression levels in parts A and B of this Figure? This would allow more direct comparison between the CBF3 over-expressors, wild type, and cbf1 mutants.

The *cbfs* mutant in Figure 4A is in *FRI* Col background, while *pSuper:CBF3-myc* line shown in Figure 4B is in Col background. It is already known that the presence/absence of a functional *FRI* allele affect the basal *COOLAIR* level (Swiezewski et al., 2009). Hence, *COOLAIR* levels among three lines were not directly comparable.

3. Figure 5. Do the authors have a qPCR assay for the various COOLAIR transcripts rather than the semi-quantitative gel-based assay used in this figure?

Yes, we performed RT-qPCR assays for the *COOLAIR* variants expressed in *FLC_ΔCOOLAIR_* lines. However, as *COOLAIR* levels in the *FLC_ΔCOOLAIR-1_* and *FLC_ΔCOOLAIR-3_* lines were too low to reliably detect by RT-qPCR and did not increase by 14-d cold treatment, we presented the data as DNA gel images. Instead, we have added the RT-qPCR results for *CAS* and unspliced *FLC* levels in *FLC_ΔCOOLAIR_* lines as shown in Figure 5—figure supplement 2.

4. Figure 5. The conclusions of the deletion analysis would be stronger if specific deletions of the actual CRT/DRE elements were made, this is likely possible with gene editing techniques.

Thank you for the valuable suggestion. We agree that the specific mutations of CRT/DREs would provide stronger evidence to our claims. However, we were unable to generate the mutant and test its vernalization response on time. Instead, the result of luciferase assays using the reporter with specific mutations in CRT/DREs shown in Figure 5A may partially address the concern.

5. I wasn't sure that Figure 8 really added anything to the paper as there isn't any new information on the actual function of COOLAIR.

Figure 8 is presenting a model of how cold-induced CBFs activate *COOLAIR* during vernalization, summarizing all the results shown in the paper. Of course, this figure does not provide any additional information, as you pointed out. But we believe this will enable a broad range of readers to understand this paper more clearly.

6. Given the differences in responses observed between this study and other it would have been informative if experiments (particularly to measure FLC and flowering time) had been carried out in different growth conditions such as in common gardens or cabinets programmed to mimic natural conditions. This may have been helpful in identifying conditions where COOLAIR transcription is required or reinforced the conclusion that it is not an essential component of the vernalization response.

We appreciate this insightful comment. As we briefly described in the Discussion section of the manuscript, it would be worthwhile to reexamine the role of *COOLAIR* for vernalization by carrying out the experiments under a natural condition or natural environment-mimicking conditions. However, under our current situation, such environmental conditions cannot be implemented. We are sorry but we have to leave it as a future interest.